Manuscript prepared for Biogeosciences
with version 2015/09/17 7.94 Copernicus papers of the LaTeX class copernicus.cls.
Date: 7 July 2016

# Underestimation of boreal soil carbon stocks by mathematical soil carbon models linked to soil nutrient status

B. Ťupek[1], C. A. Ortiz[2], S. Hashimoto[3], J. Stendahl[2], J. Dahlgren[4], E. Karltun[2], and A. Lehtonen[1]

[1]Natural Resources Institute Finland, P.O. Box 18, 01301 Vantaa, Finland
[2]Swedish University of Agricultural Sciences, P.O. Box 7014, 75007 Uppsala, Sweden
[3]Forestry and Forest Products Research Institute, Tsukuba, Ibaraki 305-8687, Japan
[4]Swedish University of Agricultural Sciences, Skogsmarksgränd, 90183 Umeå, Sweden

*Correspondence to:* B. Ťupek (boris.tupek@luke.fi), A. Lehtonen (aleksi.lehtonen@luke.fi)

**Abstract.** Inaccurate estimate of the largest terrestrial carbon pool, soil organic carbon (SOC) stock, is the major source of uncertainty in simulating feedback of climate warming on ecosystem-atmosphere carbon dioxide exchange by process based ecosystem and soil carbon models. Although the models need to simplify complex environmental processes of soil carbon sequestration, in a large mosaic of environments a missing key driver could lead into a modelling bias in predictions of SOC stock change.

We aimed to evaluate SOC stock estimates of process based models (Yasso07, Q, and CENTURY soil sub-model v. 4) against massive Swedish forest soil inventory dataset (3230 samples) organized by recursive partitioning method into distinct soil groups with underlying SOC stock development linked to physicochemical conditions.

For two thirds of measurements all models predicted accurate SOC stock levels regardless the detail of input data e.g. wheather they ignored or included soil properties. However, in fertile sites with high N deposition, high cation exchange capacity, or moderately increased soil water content, Yasso07 and Q models underestimated SOC stocks. In comparison to Yasso07 and Q, accounting for the site specific soil characteristics (e. g. clay content and topsoil mineral N) by CENTURY improved SOC stock estimates for sites with high clay content, but not for sites with high N deposition.

Our analysis suggested that the soils with poorly predicted SOC stocks, as characterized by the high nutrient status and well sorted parent material, indeed have had other predominat drivers of SOC stabilization lacking in the models presumably the mycorrhizal organic uptake and organo-mineral stabilization processes. Our results imply that the role of soil nutrient status as regulator of organic matter mineralization has to be re-evaluated, since correct SOC stocks are decisive for predicting future SOC change and soil $CO_2$ efflux.

## 1 Introduction

In spite of the historical net carbon sink of boreal soils, 500 Pg of carbon since the last ice age
(Rapalee et al., 1998; DeLuca and Boisvenue 2012; Scharlemann et al., 2014), boreal soils could
become a net source of carbon dioxide to the atmosphere as a result of long-term climate warming
(Kirschbaum 2000; Amundson 2001). They have the potential to release larger quantities of car-
bon than all anthropogenic carbon emissions combined (337 Pg) (Boden et al., 2010). In order to
preserve the soil carbon pool and to utilize the soil carbon sequestration potential to mitigate anthro-
pogenic $CO_2$ emissions, mitigation strategies of climate forcing aim to improve soil organic matter
management (Schlesinger 1999; Smith 2005; Wiesmeier et al., 2014).

Supporting soil management decisions requires an accurate quantification of spatially variable soil
organic carbon (SOC) stock and SOC stock changes (Scharlemann et al., 2014). The initial level of
SOC stock is essential in order to estimate SOC stock changes (Palosuo et al., 2012, Todd-Brown
et al., 2014), especially when estimating carbon emissions due to land-use change e.g. afforestation
of grasslands (Berthrong et al., 2009). Process-oriented soil carbon models like CENTURY, Roth-C,
Biome-BCG, ORCHIDEE, JSBACH, ROMUL, Yasso07 and Q are important tools for predicting
SOC stock change, but there are also risks for poor predictions (Todd-Brown et al., 2013, DeLuca
and Boisvenue 2012). The models need further validation and improvement as they show poor spatial
agreement on fine scale and moderate agreement on regional scale against SOC stock data (Todd-
Brown et al., 2013; Ortiz et al., 2013). Despite the potentially quantitative importance of $CO_2$ emis-
sions the expected change will be small in relation to the SOC stock. Therefore, the uncertainty
of measurements and/or model estimates could prevent conclusions on SOC stock changes (Palosuo
et al., 2012; Ortiz et al., 2013; Lethonen et al., 2015a) especially for the soils with largest SOC stocks
which are the most sensitive to carbon loss. Beside large uncertainties, the poor agreement between
the modelled and measured SOC stocks (Todd-Brown et al., 2013) could also indicate missing biotic
or abiotic drivers of long-term carbon storage (Schmidt et al., 2011; Averill et al., 2014).

For example ignoring the essential role of soil nutrient availability in ecosystem carbon use ef-
ficiency (Fernández-Martínez et al., 2014) could lead to missing important controls of plant litter
production and soil organic matter stabilization mechanisms. Soil nutrient status is linked to the
mobility of nutrients in the water solution (Husson et al., 2013), production, quality and microbial
decomposition of plant litter (Orwin et al., 2011), and formation of the soil organic matter (SOM).
The SOM affects soil nutrient status by recycling of macronutrients (Husson et al., 2013), and water
retention and water availability (Rawls et al., 2003).

In spite of state of the art soil carbon modelling based on the amount and quality of plant litter
"recalcitrance", affected by climate and/or soil properties as in the Yasso07, Q and CENTURY mod-
els, these type of process based models do not include mechanisms for SOM stabilization by a) the
organic nutrient uptake by mycorrhizal fungi; b) humic organic carbon interactions with silt-clay
minerals; and c) the inaccessibility of deep soil carbon and carbon in soil aggregates to soil biota

(Orwin et al., 211; Sollins et al., 1996; Torn et al., 1997; Six et al., 2002; Fan et al., 2008; Dungait et al., 2012; Clemente et al., 2011). Although the models do not contain aforementioned mechanisms and controls for changes in SOM stabilization processes, they have been parameterized using a wide variety of datasets and can treat soil biotic, physicochemical and environmental changes implicitly. The Yasso07 model (Tuomi et al., 2009, 2011) is an advanced forest soil carbon model and it is used

for Kyoto protocol reporting of changes in soil carbon amounts for the United Nations Framework Convention on Climate Change (UNFCCC) by European countries e.g. Austria, Finland, Norway, and Switzerland. The Q model (Ågren et al., 2007) is a mechanistic litter decomposition model developed in Sweden and used e.g. to compare results produced with Swedish national inventory data (Stendahl et al., 2010, Ortiz et al., 2011) and also with other models at national or global scales

(Ortiz et al., 2013; Yurova et al., 2010). The CENTURY model (Parton et al., 1987, 1994, Adair et al., 2008) is one of the most widely applied models and it is used for soil carbon reporting to UNFCCC by Canada, Japan, and USA. Although individual parameters and functions vary, mathematical models such as Yasso07, Q and CENTURY have similar structures. For example, these models are driven by the decomposition rates of litter input and soil organic matter (SOM). Decom-

posing litter and SOM is divided into pools based on litter quality, and its transfer from one pool to another is apart from model functions and parameters affected by temperature (Q) and/or water (Yasso07), and/or soil texture and structure (CENTURY). The Q model does not include explicit moisture function, whereas for the Yasso07 and CENTURY models precipitation affects decomposition (Tuomi et al., 2009; Adair et al., 2008). On the other hand, the models do not explicitly or by

default include mechanisms that reduce decomposition by excessive precipitation/moisture (Falloon et al., 2011).

We hypothesized that (1) soil carbon estimates of the Yasso07, Q, and CENTURY models would deviate for soils where SOC stabilization processes not implicitly accounted by the models are predominant, (2) the Yasso07 and Q models ignoring soil properties would fail on the nutrient rich sites

of South-West coast of Sweden and on occasionally paludified clay and silt soils, and (3) the CENTURY model outperforms the Yasso07 and Q models due to fact that it includes soil properties as input variables.

We grouped Swedish forest soil inventory data into homogenous groups with specific soil physicochemical conditions using regression tree and recursive partitioning modelling methods. After that

we ran the models into an equilibrium with a litter input which was derived from the Swedish forest inventory. Thereafter we compared the model estimates against data by groups that were obtained from the regression tree model. In discussion we address the reasons why the models deviate and indicate directions of further improvements.

## 2 Material and methods

 ## 2.1 Measurements

We analysed data from the Swedish forest soil inventory (SFSI) which is a stratified national grid survey of vegetation and physicochemical properties of soils (SLU, 2011, Olsson et al., 2009). All analysis was done using R software for statistical computing and graphics (R core team 2014). The soil data were identical to dataset used in Stendahl et al. (2010). We restricted our sample plots to minerogenic soils since the Q, Yasso07, and CENTURY models were not developed for use on peat soils, and only to plots for forest land use with Swedish forest inventory data (SFI). We also excluded samples with total SOC stock below 2.8 and above 470.5 ($tC\,ha^{-1}$), i.e. samples with SOC stock below 0.01 and above 99.9 percentile. Measurement data originated from the 1993 to 2002 which constitute a full inventory, and from 2020 sample plots located around Sweden, and in total it including 3230 samples. For each sample plot the weather (years 1961-2011) and N deposition (years 1999-2001) data was retrieved from the nearest stations of Swedish Meteorological and Hydrological Institute (SMHI) network (Fig. 1). The plots which were linked by the closest distance to the given weather station had the same weather and N deposition data, and the number of soil samples per station ranged between 10 and 70. The mean total SOC stock of samples corresponding to weather stations ranged between 40 and 200 ($tC\,ha^{-1}$), and the SOC stock level increased from the South to North of Sweden (Fig. 1).

Each sample plot contained categorical data from the field survey on the sorting of soil parent material, humus type, soil texture, and soil moisture. In our analysis we reduced categorical classes by basing them on the sorting of soil parent material and humus type (Table 1). We determined numeric values for silt, clay, and sand content from soil texture categories by Albert Atterberg's distribution of the different grain size fractions in tills and by Lindén's (2002) distributions for sediments (Table 1). We also determined numeric values of volumetric soil water content (SWC) from categorical field data classified according to the depth of the ground water level (WL) (Table 1).

As typical for soil carbon inventories, the variation of data was large (Table 2). For example, the mean total SOC stock of all samples was 93 ($tC\,ha^{-1}$) while 1st and 99th percentiles were 17 and 309 (Table 2). The mean SOC stock was 33.3 and 66.8 ($tC\,ha^{-1}$) for the humus horizon and the mineral soil. The mean values of cation exchange capacity (CEC ) 23.9 ($mmol_c\,kg^{-1}$), the base saturation 36.4%, and the C/N ratio 16.5 indicated conditions of medium fertility, although the soils were mostly acidic (mean pH was 5.2). The mean prevailing soil water content (22.3) was typical for the well-drained forest soils. The mean annual temperatures ranged from below 0 to above 8 °C, and annual precipitation varied between 392 and 1154 mm (Table 2). Total SOC stock for all the samples generally increased for peat and peat like humus forms, for well sorted sediments, for soils with high fraction of silt and clay and with increasing soil moisture (Fig. S1).

### 2.1.1 Biomass and litterfall estimates

For the biomass and litterfall estimation we adopted standard method of national greenhouse gas inventories for estimating soil carbon stock changes (Statistics Finland 2013). In order to model SOC stocks of forest in equilibrium (not SOC stocks changes) we modified the methos by estimating the long term litterfall of forest in equilibrium. Forest stand biomass was estimated by allometric biomass functions for stem with bark, branch, foliage, stump, coarse-roots and fine-roots applied to basic tree dimensions (breast height diameter, total height of tree, number of trees) of SFI stands (Marklund 1988; Pettersson and Ståhl 2006; Repola 2008; Lehtonen et al., 2015b). In order to simulate "equilibrium" soil carbon stock we estimated long term mean forest biomass, referred to as "equilibrium forest" below.

We adopted an observed fraction of photosynthetically active absorbed radiation ($f_{APAR}$, Fig. A1) as a relative indicator of a site's capacity to produce biomass (minimum = 0, maximum = 1) by accounting for the forest stand structure, ranging from the absent stand $f_{APAR} = 0$ to the closed canopy stand $f_{APAR} = 1$, through its major role on limiting of the potential gross primary production (Peltoniemi et al. 2015). The $f_{APAR}$ was calculated based on SFI measurements of basic tree dimensions as in Härkönen et al. (2010) and for the main tree species (pine, spruce, deciduous) it was well correlated with the stand basal area (Appendix A).

The equilibrium forest $f_{APAR}$ values were assumed to be in a range between the median and the maximum fraction of the observed state forest $f_{APAR}$ for a given species, latitudinal degree, and site productivity class (Appendix A). We selected equilibrium $f_{APAR}$ as the 70th percentile ($f_{APAR70}$) out of a range from the 50th to 95th, because the modelled soil carbon distributions with a litter input from the $f_{APAR70}$ biomass best agreed with the measured soil carbon distributions (Fig. S2). The $f_{APAR70}$ was the estimated 70th percentile of the observed fraction of absorbed radiation specific for a given species, latitudinal degree, and site productivity index H100 (height of trees at 100 years of age, m, Hagglund and Lundmark 1977) (Fig. B1). Instead of modelling of equlibrium biomasses for every tree stand component separately for the species, latitude, and site productivity index, we simplified the biomass modelling firstly by estimating only equlibrium forest stand structure for the species, latitude, and productivity ($f_{APAR70}$, Table A1) and secondly by using $f_{APAR70}$ with $f_{APAR}$ biomass models (Table B1) to estimate the biomass components.

We modelled the equilibrium biomass by applying the fitted exponential functions between the observed state forest biomass components (stem, branch, foliage, stump, coarse-roots, fine-roots, estimated by tree stand measurements and the allometric biomass functions) and the observed fraction of absorbed radiation ($f_{APAR}$) (Appendix B) to the estimated $f_{APAR70}$ of the equilibrium forest. The understory vegetation of the equilibrium forest was estimated by applying our ground vegetation models (Appendix C) to the modelled equilibrium forest characteristics, and plot specific environmental conditions.

In order to derive the litter inputs, annual turnover rate (TR, the fraction of living biomass that is shed onto the ground per year, unitless) of biomass components were applied to the modelled biomass components of the equilibrium forest. The needle litter TR was a linear function of latitude for pine and spruce and a constant for deciduous species (Ågren et al., 2007). The TR of branches and roots were from Mukkonen and Lehtonen (2004), Lehtonen et al. (2004) and the TR of stump and stem were from Viro (1955), Mälkönen (1974, 1977) as cited in Liski et al. (2006). For tree fine roots we assumed there was a difference between tree species and between southern and northern Sweden. For pine, spruce, and birch the fine roots TR were 0.811, 0.868, and 1.0 respectively as reported by Maidi (2001) and Kurz et al. (1996), and cited in Liski et al. (2006). Kleja et al. (2008) and Leppälampi-Kujansuu et al. (2014) reported different fine root TR for Southern (1 and 0.83) and Northern Finland (0.5). We interpolated TR according to the mean annual temperature gradient between TR of fine roots in the South and the North. The fine roots TR of 0.811, 0.868, and 1.0 in the warmest southernmost soil plots were thus reduced down to 0.5 in the coldest northernmost soil plots. The understory TR were applied as in Lehtonen et al. (manuscript).

The major part of the litter input originated from the tree stand biomass components which were modeled by the non-linear functions with $R^2$ values close to 0.9 (Fig. B1, Tables A1 and B1). The linear understory vegetation models had low $R^2$ values (Table C1). However, when the understory models (Appendix C) were applied only to plots close to equilibrium forest, as in our application, the $R^2$ values of predicted and observed understory components were larger (Fig. S9). In comparison to major understory litterfall originating from reasonably well predicted dwarf-shrubs and mosses (Fig. S9 and S10), the influence of poorer understory models (for herbs, grass, and lichens) was small on predictions of the understory litter and marginal on predictions of the total forest litterfall (Fig. S10). The main improvement on the accuracy of total litter input was achieved by avoiding the confounding effect of observed forest state by modelling the biomass/litterfall estimates representing the mean long-term conditions (defined by estimated equilibrium $f_{APAR70}$) for small regions (defined by degree of latitude and productivity class for dominant species, Fig. A1). Thus the estimates accurately reflected the long-term spatial variability in dominant species, nutrient status and climate (Fig. S11) and lacked higher spatial and temporal precission; as attempts for high precision of the estimates applied for the period of the last few thousands of years would be uncertain due to high variation of factors affecting plot history.

### 2.1.2 Correlation analysis

Overall our data consists of 3230 soil samples and their carbon stocks linked to soil physicochemical variables, stand and ground vegetation biomass and litterfall components, and nearest weather station environmental variables. We performed the Spearman's rank correlation analysis between the total soil carbon stock and the other soil variables, site, climate and vegetation characteristics. As expected the total soil carbon stock most strongly correlated with the measured variables used for its

calculation e.g. bulk density, depth of humus and mineral soil, carbon content, and stoniness. These variables were excluded from further regression tree analysis which aimed to group data according to the processes of soil carbon stock development.

### 2.1.3 Regression trees

In order to organize SOC data into groups according to the physicochemical soil variables and to better understand the nature of measured data, we generated regression trees of SOC stocks by using recursive partitioning (RPART) (Therneau and Atkinson 1997). RPART is based on developing decision rules for predicting and cross validation of continuous output of soil carbon stocks (regression tree). The classification tree was built by finding a single variable which best splits the data into two

groups. Each sub-group was recursively separated until no improvement could be made to the soil carbon stock estimated by using the split based regression model. The complex resultant regression tree model was cross validated for a nested set of sub trees by computing the estimate of soil carbon stock to trim back the full tree.

When building the regression tree models we excluded variables such as bulk density, carbon

contents of soil layers, soil depth, and stoniness, since these measured variables were used for determining the total soil carbon stock. The selected variables for the RPART data mining were based on the correlations analysis (see 2.1.2), the processes of soil organic matter formation (e.g. Husson et al., 2013) and decomposition, and represented the soil categorical variables (sorting of parent material, soil texture, long-term soil moisture and humus form), soil physicochemical variables (sand,

clay, and silt content, long-term soil moisture, highly bound water, C/N ratio, pH, CEC of organic, B, BC, and C horizons), climatic variables (annual mean air temperature, annual precipitation sum), and stand and site characteristics (tree species coverage of pine, spruce and deciduous, total foliar litter input, productivity class and N deposition). Alternatively we also ran regression and classification analysis by excluding all measured soil variables because soil variables are often unavailable

for landscape level modelling.

The regression tree model separated the measured total SOC stocks ($tC\,ha^{-1}$) into 10 groups. The cation exchange capacity of the BC horizon (CEC, $mmol_c\,kg^{-1}$) divided all the samples into 2/3 of lower SOC stock groups (means between 65 and 130 $tC\,ha^{-1}$) and 1/3 of larger groups (means between 86 and 269 $tC\,ha^{-1}$) (Fig. 2a). The group of the smallest SOC stock consisted of

230 959 samples compared to 8 samples of the group with the largest SOC stocks. We acknowledge that this is a small distinct group based only on 8 observation. However, we did not have any reasons to exclude these datapoints as outliers. Two-thirds of samples with smaller SOC stocks were subdivided by CEC and the type sorting of soil parent material (sorted or unsorted). One-third of samples with larger SOC stocks was subdivided by the C/N ratio, CEC, N deposition among others. Roughly

generalized, groups from left to right or from 1 to 10 formed a gradient in levels of SOC stock, moisture, nutrient status, and production (Fig. 2, Table S1).

The alternative regression tree model was built with variables other than soil properties. The regression tree with the annual mean air temperature, the annual precipitation sum and the percentage of pine trees in the stand, and the nitrogen deposition separated measured SOC stocks ($tC\,ha^{-1}$) into five groups (Fig. S3). Colder groups with smaller SOC stocks (means 67 and 85) also had less litter input (below $3\ tC\,ha^{-1}$) and low site productivity index (height of dominant trees at 100 years of age, H100 < 20 m) (Table S2). The site index H100, that can be translated to a specific productivity $m^3\,ha^{-1}\,yr^{-1}$, was calculated for sites based on observed site properties from Swedish forest inventory by using the methodology of Hägglund and Lundmark (1977) (Swedish Statistical Yearbook of Forestry 2014). Nitrogen deposition only slightly impacted the higher productivity class of soils and litter input (Table S2).

## 2.2   Soil carbon stock modelling

The Q model (Rolff and Ågren, 1999) is a continuous mechanistic litter decomposition model describing change of soil organic matter over time. The decomposition rate for the branch, stem, needle, fine root, and woody litter fractions is controlled by the temperature, litter quality, microbial growth and litter invasion rate. The model has been calibrated for seven climatic regions of Sweden in order to account for Swedish temperature and precipitation gradients (Ortiz et al., 2011) (Table 3). The Q model was applied in several studies of SOC stock and change estimation in Sweden (e.g. Stendahl et al., 2010; Ortiz et al., 2013; Ågren et al., 2007). The Q model was run for seven Swedish climatic regions (Ortiz et al., 2011). The mean regional parameterization from the calibration of the 2011 Q model was used for the plot simulations. Thus, the simulations in each region represent variations in climate and litter input and not parameter variations. The equilibrium soil carbon stocks are estimated in the model using the equation for equilibrium soil carbon stock which is derived from the decomposition functions with constant amounts and quality of litter input.

The Yasso07 model (Tuomi et al., 2009; 2011) is one of the most widely applied SOC models. The model was calibrated based on almost 10 000 measurements of litter decomposition from Europe, North and South America (Table 3). The required annual inputs of litterfall, its size and chemical composition, temperature and precipitation determine the decomposition and sequestration rates of soil organic matter. Yasso07 estimates SOC stock to a depth of 1 m (organic and mineral layers), change of SOC stock, and heterotrophic soil respiration. Species specific chemical composition of different litter compartments of Yasso07 were used according to Liski et al. (2009). The initial soil organic matter of Yasso07 was zero. The simulated soil carbon stock corresponding to equilibrium between the litter input and decomposition was achieved by a Yasso07 spin-up run of 10 000 years. Yasso07 runs used litter inputs of the equilibrium forest biomasses (see 2.1.1) and climate variables (annual air temperature, monthly temperature amplitude, and annual precipitation). The global parameter values of decomposition rates, flow rates, and other dependencies of Yasso07 soil carbon model were adopted from Tuomi et al. (2011) and the estimates of Yasso07 SOC stocks were used

in comparison with measurements and other models. We did not use the SOC stocks simulated with the more recent Yasso07 parameters based on the litter decomposition data from the Nordic countries (Rantakari et al., 2012), because the SOC stocks simulated with the global parameter values produced better fit with SFSI measurements.

The CENTURY mathematical model originally developed for grassland systems (Parton et al., 1987; 1992) has been since modified for various ecosystems including boreal forests (Nalder and Wein 2006). The CENTURY is also one of the most widely applied models. The soil organic matter in the model consists of active, slow, and passive pools which have different TR (Table 3). The decomposition rates are modified by temperature and moisture, and in addition the decomposition rates of the slow and passive pools rely on lignin to N and C to N ratios, while the active pool decomposition rate relies on soil texture. The model simulates soil organic matter to a depth of 20 cm. The model simulates plant production and pools of living biomass, while TR for biomass pools determine the litterfall inputs to soil. To compare the performance of the soil sub-model with other soil carbon dynamics models, Q and Yasso07, we only used the CENTURY soil sub-model. We used the same litterfall inputs as used by the Q and Yasso07 simulations, which were estimated by our litterfall modelling (see 2.1.1). The litter inputs reflected N deposition and site productivity (Fig. S11). For CENTURY we adopted general parameters from the parameter file "tree.100", parameters of site "AND  H_J_ANDREWS" for conifers, and site "CWT  Coweeta" for deciduous trees. The N dynamics in CENTURY sub-model included tuning site specific parameters of topsoil mineral N relative to N deposition (Throop et al. 2004) and reduction of C/N ratio of the litterfall up to 15% for most productive sites (Merilä et al. 2014). We also accounted for site specific soil drainage by varying its parameter between 1 and 0.6 relative to long-term soil water content ranging between 10 and 50% (Raich et al. 2000). The CENTURY SOC stocks simulation were run with equilibrium forest litter inputs, site specific C/N ratios of litterfall, site specific soil parameters (specific bulk density, sand, silt, and clay content, mineral N in topsoil, and drainage) and climate variables (monthly air temperature, and monthly precipitation). In order to account for the deep soil carbon (Jobbágy and Jackson 2000), we scaled CENTURY estimates representing the topsoil horizon by adding 40% of estimated site specific SOC stock. The simulated equilibrium SOC stocks were estimated by a spin-up run of 5 000 years. The number of years to reach equilibrium (equilibrium between the litter input and decomposition) was sought empirically on 100 random sites, and differs from Yasso07 and Q models.

## 3   Results

The distributions of Yasso07, Q, and CENTURY model estimates of total SOC stocks ($\mathrm{tC\,ha^{-1}}$) were in agreement for 2/3 of the measured data with lower SOC stock (Fig. 3, distributions of groups 1, 2, and 4). The remaining 1/3 of data was underestimated by models. This 1/3 of data was

separated into 7 physicochemical soil groups (means of groups in range from 104 to exceptionally large 269 $tC\,ha^{-1}$, see Fig. 3, distributions of groups 3, and 5-10). The linear regression of mean levels of all 10 physicochemical soil groups (weighted by the number of samples in each group) between the modelled and measured SOC stocks showed smaller underestimation of CENTURY compared to Yasso07 and Q models (Fig. 4). The weighted root mean square error (RMSE) was 27.5 $(tC\,ha^{-1})$ for CENTURY and 31.6 and 38.8 for Yasso07 and Q respectively. The proportion of explained variance was larger for Q ( $r^2$ = 0.58) than for Yasso07 and CENTURY ( $r^2$ = 0.42 and 0.32) (Fig. 4). The deviation of the distributions of CENTURY SOC stocks, simulated using soil bulk density, sand, silt, and clay content, were lower than those of Yasso07 and Q estimates for 10 physicochemical soil groups (Fig. 3). Accounting for site specific soil texture (clay, silt, and sand content) and structure (bulk density) by CENTURY model improved SOC stock estimates for fertile sites with high clay content, but not for sites with high N deposition. Varying CENTURY parameters of site specific topsoil mineral Nitrogen and C/N ratio of the litterfall showed that this impact on SOC stocks estimates was small in comparison to sensitivity of SOC stock estimates to litterfall (Fig. S12). The application of site specific drainage on our mostly well drained soils showed minor impact on estimated CENTURY SOC stocks.

As expected, the models clearly showed less variation than the measurements. The shift of the mean values from the center of distribution, the width of confidence intervals of means, and the width of the tails of distributions were clearly larger for the measurements than for the modelled estimates (Fig. 3). The modelled distributions agreed for the poor-medium fertility soils with low and medium measured SOC stocks, low and medium CEC, unsorted parent material, low temperatures and low production (groups 1, 2, and 4) (Fig. 2, Table S1, Fig. 3). Disagreement between modelled and measured SOC stock distributions were formed on fertile soils with sorted parent material (groups 3 and 5), soils with higher water content (groups 3, 5, and 10), where nitrogen deposition was large (groups 7 and 8), and where CEC was median or large (Fig. 2, Fig. 3). The largest deviation between the measured and modelled distributions was found for the relatively small physicochemical groups of soils (3%) typical for highly bound water and peat humus types (groups 8 and 10) (Fig. 2, Fig. 3). The distributions of measured total SOC stocks $(tC\,ha^{-1})$ generally increased for the groups with higher nutrient status (Fig. 3, Fig. S4). The distributions of SOC stocks in mineral soil were larger than those in humus horizon, and distributions of mineral SOC stocks increased with fertility slightly more than distributions of SOC stocks in humus horizon (Fig. S4).

After excluding all the soil physicochemical characteristics from the recursive partitioning, the SOC stock distributions of 5 groups regression tree model (Fig. S3, Table S2) were in agreement between the measurements and model estimates for 3 groups (77% of samples) and deviated for 2 groups (23%) (Fig. S5). The modelled SOC stock distributions agreed with measurements for all models on sites with low annual temperatures < 3 °C in northern sites (low-C.cold.pine, low-C.cold.other) and for warmer conditions in middle Sweden on sites with low nitrogen deposition

and median SOC stocks (Fig. S5). However, the models underestimated SOC stocks on sites with high ($> 10 \, \mathrm{kgN \, ha^{-1} \, y^{-1}}$) N deposition (21% of samples) and on sites with warm and dry climate (2% of samples) (Fig. S5).

The variation of density functions of modeled SOC stocks for 10 physicochemical groups (Fig. 3) was similar to the variation of the total annual plant litter input ($\mathrm{tC \, ha^{-1}}$) (Fig. S6) indicating that
litterfall was the main driver of SOC accumulation in the models . The mean levels of annual plant litter input and mean SOC stocks for 10 soil groups were more strongly correlated for Yasso07 and Q models (with $r^2$ values 0.86 and 0.96, respectively) than for CENTURY ($r^2 = 0.52$). Although, models performed reasonably well for the largest soil groups of nutrient and production levels (Fig. 3 and Fig. 4), none of the models was able to predict variation of individual samples (Fig. S7). The
model estimates were well correlated between Yasso07 and CENTURY with $r^2$ ranging from 45 to 73% for individual samples of 10 soil groups, whereas the correlations of estimates between Q and the other two models were lower (Fig. S8).

## 4 Discussion

### 4.1 SOC stock distributions linked to mechanisms of SOM stabilization

It has been suggested that process based soil carbon models with the current formulation lacking major soil environmental and biological controls of decomposition would fail for conditions where these controls predominate (Schmidt et al., 2011; Averill et al., 2014). Although, the effect of the soil properties on SOC stocks e.g. soil nutrient status in the widely used models such as Yasso07, Q, and CENTURY have not previously been quantitatively evaluated. We found that in comparison
with Swedish forest soil inventory data, the models based on the amount and quality of inherent structural properties of plant litter (Q, Yasso07, and CENTURY) produced accurate SOC stock estimates for 2/3 of northern boreal forest soils in Sweden. Two-thirds of the distributions of SOC stocks measurements of SFSI agreed with distributions of SOC stock estimates of the Q, Yasso07, and CENTURY soil carbon models (Fig. 3, distributions of groups 1, 2, and 4). However, the SOC
stocks underestimation by these models for one third of the data (Fig. 3, distributions of groups 3, and 5-10) indicated that some drivers other than molecular structure, especially site nutrient status, play an important role in higher SOC stocks sequestration.

Some level of deviation from measurements and poorly explained spatial variation (Fig. S7) was expected from the uncertainties of the SOC measurements, annual plant litter inputs and climate
variability for the model SOC stock change estimates (Ortiz et al., 2013; Lehtonen et al., 2015a). For the long-term SOC stock development the model uncertainties are less known than for the short-term litter decomposition. Previously reported fine scale comparison also showed poor agreement between Earth system models and the Northern Circumpolar Soil Carbon Database (Todd-Brown et al., 2013), although drivers of the deviation still remained open. Our results showed that if mod-

els strongly depend on the litter inputs (Fig. S6) then the spatial differences between measured and modeled SOC stock distributions could be linked to sites with rich nutrient status through cation exchange capacity, C/N ratio, N deposition, drainage (sorting of parent material) among other factors (Fig. 2 and 3). Additionally, when the soil properties were excluded from the regression, the estimates of SOC stocks also deviated for the fertile groups (Fig. S5). However, the rich nutrient status for these groups was linked to differences in species composition, N deposition, and climate (temperature, precipitation) instead of soil properties (Fig. S3).

Larger net soil carbon accumulation in nutrient rich sites could be attributed to the relative differences in litterfall components (relatively more leaves and branches with higher N content than fine roots), and to higher N availability and carbon use efficiency of decomposers, reduction of respiration per unit of C uptake (Ågren et al., 2001, Manzoni et al., 2012, Fernández-Martínez et al., 2014). Largest deviation between measured and modeled data in our study was found for fertile presumably N rich and fresh to fresh-moist sites. The soils with large N deposition were also highly productive and showed high to exceptionally high SOC stocks (Fig. 2, Fig. 3, soil groups 7 and 8). This was in agreement with fertilization and modelling study of Franklin et al. (2003) showing an increase in soil C accumulation with N addition. Our forest biomass and litterfall estimates were based on forest inventory and modeling, but the site nutrient status and N deposition was only partially reflected in the amount of biomass/litterfall (Fig. S11) and its quality. The quality was only reflected through the biochemical differences between species and plant litter components. The relative differences between the biomass/litterfall components or between C/N ratios of litterfall in relation to site fertility are not accounted by the current biomass models, but soil fertility could be considered in an attempt of SOC stock modelling (included in CENTURY but missing in Yasso07 and Q models). For example the proportion of acid -, water -, and ethanol-soluble and non-soluble litter inputs for Yasso07 could be re-evaluated by allowing it to vary depending on site fertility, in addition to currently used variation specific for species and the litter components. Although CENTURY SOC stocks were sensitive to the amount of clay, the variation of topsoil mineral N and C/N ratio of litterfall did not improved SOC stock predictions for sites with high N deposition (Fig. 3 and Table S1).

The litter decomposition and SOC stabilization rates in Yasso07, Q, and CENTURY based on the litter quality "recalcitrance" originating from the litter bag mass loss measurements have major drawbacks. The mass loss from the litter bags is assumed to be fully mineralized, although the litterbags are subjected to non-negligible leaching (Rantakari et al., 2012; Kammer and Hagedorn, 2011). The SOC stabilization represented in models by the remaining litter mass is thus underestimated due to the fraction of particulate organic matter and dissolved organic carbon that is lost from the litterbags but later immobilized e.g. through organo-mineral stabilization. The use of stable isotopes seems to determine the field carbon mineralization and accumulation rates from the labile (high C quality and N concentration) or recalcitrant (low C quality and N concentration) litter more accurately than litter bags (Kammer and Hagedorn, 2011).

Higher amount of more recalcitrant fine roots compared to more labile leaves (Xia et al., 2015) heavily increased the soil carbon sequestration in CENTURY model simulations which was in line with McCormack et al. (2015). Though, the contribution of fine roots to SOC stabilization is still not settled due to the significant role of mycorrhizal fungi in SOC accumulation (Averill et al., 2014; Orwin et al., 2011). Xia et al. (2015) claimed that more recalcitrant fine roots contribute to stable SOC more than leaf litter, because fine roots degrade slower. This would be supported by the fact if the precursors of fine roots that are degraded by fungi are more stable than the precursors of leaves degraded by microbes. However, more recalcitrant plant litter has been also suggested to stabilize less SOC stocks (Kammer and Hagedorn, 2011). This is a result of recalcitrant litter satisfying less of the microbial N demands promoting respiration and reducing the long-term production of microbial products, precursors for the organo-mineral stabilization (Cotrufo et al., 2013, Castellano et al., 2015). According to the microbial efficiency-matrix (MEM) stabilization mechanism (Cotrufo et al., 2013) fertile sites with relatively more labile plant litter, but with larger absolute production and larger microbial activity than poor sites, would in long-term stabilize more carbon through organo-mineral stabilization. Our results supported MEM stabilization theory by showing larger carbon stocks in mineral soil than in humus horizon, and by relatively more SOC stocks in mineral soil in fertile groups than in poor conditions (Fig. S4).

Expanding on the CENTURY model structure, the MySCaN model incorporating the organic nutrient uptake by mycorrhizal fungi estimated positive effect on SOC accumulation, relatively larger in poor than in fertile sites (Orwin et al., 2011). Therefore, not accounting for the organic nutrient uptake by mycorrhizal fungi by the Yasso07, Q, and CENTURY models probably led to the underestimation of SOC stocks in sites with higher nutrient status. This hypothesis needs to be tested in further studies. We did not have all input data and the source code to include MySCaN into our model intercomparison. The spatial trends of N and P data of litter in Sweden that would be needed to make such study were not available. However, adjusting biomass turnover rates, used for the litter input estimation, in dependence to site fertility would lead into larger inputs for fertile sites and increased SOC stock accumulation as a result of increasing plant productivity and inputs. It is well established that SOM increases soil fertility by improving the soil water and nutrient holding capacity; recycling of SOM increases CEC, humic substances and nutrient availability for plant resulting in larger biomass/litter production (Zandonadi et al., 2013). As an alternative to adjusting turnover rates with site fertility, we suggest that a feedback link in models between increasing fertility due to SOC stock accumulation (e.g. due to increased CEC relative to humus, increased nitrogen availability), increasing litter inputs, and reduced rates of SOC decomposition per unit of litter input (e.g. through satisfying more microbial N demand with less respiration, limited oxygen in increased moisture conditions) would also increase SOC stock accumulation.

Increased moisture and more frequent water saturation due to SOC accumulation limits soil oxygen availability and slows rates of microbial decomposition which increases the rate of SOC sta-

bilization. Our results, which were derived from mostly well drained soils, suggest that measured high SOC stocks may be partly caused by reduction of decomposition at increased water content (Fig. 2). The CENTURY model has an optional function that represents the reduction of decomposition caused by anaerobic conditions. The function becomes active when a controlling parameter, "drain", is changed, and the value of the parameter has to be arbitrarily determined through parameter fitting against SOC data (e.g. Raich et al., 2000). However, this function was meant for anaerobic conditions in poorly drained soils, therefore it was not applicable to the prevailing conditions of our sites. Accounting for drainage only on some sites slightly affected decomposition, when precipitation increased and potential evapotranspiration decreased in late spring or early autumn. Water availablility affecting soil fertility and SOC formation is beside climate also affected by topography (Clarholm et al., 2013) which was not accounted for by CENTURY. Detailed modelling of soil water conditions requires specific functions and many parameters, which are not included in simpler SOC models like Q and Yasso07. However, appropriate modelling of soil water conditions and reduction of decomposition in wet conditions (not necessarily at saturation) would potentially improve the performance of SOC models in particular for soils with high SOC stocks.

### 4.2  Intercomparison of models

The similarities between the variations of modeled SOC stocks and litterfall inputs for the soil groups with different fertilities (Fig. 3, Fig. S6) could be expected for the Yasso07 and Q models which ignore the soil properties. These models run organic matter decomposition and humus stabilization with litterfall, temperature and/or precipitations input data. Litter quality as input in Yasso07 and Q implicitly includes some information on soil properties, but as we saw litter quality hardly mapped any of soil fertility. Although, the impact of soil properties on the estimates was seen in the more complex CENTURY model for sites with high clay content, the SOC stock of sites with high N deposition were underestimated. The CENTURY model depended less on the amount of litter input, and its variations of the estimated SOC stocks distributions were less pronounced than those for the Yasso07 and Q models. In testing multiple soil carbon models with same litter inputs Palosuo et al. (2012) observed larger variation in modeled SOC stocks at the early stage of the litter decomposition (10 years) but later on at 100 years the variation decreased. Although the variations of SOC stocks were similar between the models, the estimated CENTURY SOC stocks distributions were lower than the Yasso07 estimates when we did not accounted for deep soil carbon. CENTURY in its original configuration simulated SOC stock up to 20 cm soil depth (Metherell et al., 1993) whereas the Yasso07, Q, and measured SOC stocks data represented up to 100 cm of the soil (Tuomi et al., 2009, Stendahl et al., 2010). In Yasso07 model parameters were calibrated based on soil age chronosequence data of SOC stocks for soil depths up to 30 cm, which was assumed to represent 60% of the total SOC stocks up to 100 cm soil depth (Liski et al., 1998, 2005 as cited by Tuomi et al., 2009). Therefore, when 40% of the missing deep carbon (Jobbágy and Jackson 2000) were

added on top of the original CENTURY estimates as was done when callibrating Yasso07, the SOC
stock levels for CENTURY were larger than those for the Yasso07 and Q models.

Although estimated SOC stocks of CENTURY were generally larger than those of Yasso07, the
correlation between CENTURY and Yasso07 estimates was stronger than for Q model compared to
two other models (Fig. S8). The reason was probably similar global parameterizations of Yasso07
and CENTURY whereas Q was specifically parameterized and applied for the regions in Sweden
(Ågren and Hyvönen 2003, Ortiz et al., 2013). Furthermore the Q model SOC stock estimates
were more sensitive to differences in species coverage e.g. to pine and spruce (Ågren and Hyvönen
2003) and formed two distinct point cloud distributions (one for pine and broadleaves, the other for
spruce) when compared with the CENTURY or Yasso07 estimates (Fig. S8). In spite of similarities
in Yasso07 and CENTURY SOC stocks estimates, Yasso07 was more sensitive to species cover-
age through species specific litterfall solubility (Liski et al., 2009) than CENTURY which treated
conifers in a single group (Metherell et al., 1993). Pine and other species (spruce) coverage was
shown to affect measured low and median SOC stocks of colder climate if the soil properties were
not considered (Fig. S5). Therefore the pattern of increased accumulation of SOC stock on sites
with larger spruce coverage partially observed in distribution of Yass07 estimates, and missing in
the CENTURY estimates, could be related to the slightly lower solubility/decomposability of spruce
compared to pine litterfall. However, the CENTURY model SOC stocks were also highly sensitive to
accurate estimation of fineroots litterfall (Mc Cormack et al., 2015) typically increasing with colder
climate and increasing the C/N ratio of the organic layer (Lehtonen et al., 2015b) which is driven by
the dominant tree species (Cools et al., 2014).

Large SOC stocks measurements on sites with high long-term nitrogen deposition over 10
$\mathrm{kgN\,ha^{-1}\,y^{-1}}$ (Fig. 3 and Fig. S4) were underestimated by the Q, Yasoo07, and CENTURY models.
A positive correlation between nitrogen deposition and SOC stocks measurements in Sweden had
been previously reported by Olsson et al. (2009), and the modelling study by Svensson et al. (2008)
indicated that Swedish soil carbon was decreasing in the North and increasing in the South mainly
as a result of different nitrogen inputs. The Q and Yasso07 models do not have nitrogen processes.
As for CENTURY, it is reported that large N input could enhance plant productivity and then in-
crease SOC (Raich et al., 2000). The purpose of the study was to evaluate the performance of soil
carbon models against the same SOC data using the same litter input, and therefore only the soil
carbon submodel was used and the feedback of nitrogen input to plant productivity was primarily
included in this study indirectly, through estimated equilibrium litter input based on site productivity
class which strongly correlated with N deposition (Fig. A1 and S11). In spite of slight increase of
SOC stock estimates when CENTURY accounted for the site specific topsoil mineral N, C/N ratio
of litterfall, in sites with large N deposition CENTURY still underestimated. However, as in the
case of drainage discussed above, the CENTURY incorporates more detailed processes than the rel-

atively simpler soil carbon models, Q and Yasso07, do, and hence the CENTURY could potentially reproduce a wider range of SOC stocks if it was parameterized with more detailed data.

## 5 Conclusions

In this study we presented the reasons to re-evaluate the connection between the soil nutrient status and performance of widely applied soil carbon models (Yasso07, Q, and CENTURY). As previously described in detail, our simulation was based on the widely used process based SOC models, accurate driving data including litter inputs, and massive SOC data points (Swedish inventory data, N=3230). The models differed in the main controls and functions and their performance was expected to depend on model complexity (CENTURY outperforming Q and Yasso07). The intercomparison of SOC stocks between Yasso07, Q, and CENTURY models and Swedish soil carbon inventory data revealed that these process based mathematical models developed for predicting short-term SOC stock changes can all in their current state predict accurate long-term SOC stocks for most soils. The estimates of CENTURY fitted generally better to measurements than those of Yasso07 and Q model. However, Yasso07 model which requires fewer parameters and less input data showed similar performance than CENTURY, except for sites with hig clay content. The models with their current formulation lack nutrient status related controls of decomposition and soil carbon accumulation and underestimated for conditions where the high nutrient status predominate, in our application for medium-highly productive sites of Southern Sweden.

Through the intercomparison of three different widely-used SOC models with massive data points, we identified that re-evaluating of the impact of nutrient status would improve the model development towards their accuracy. Particularly, the relationship between the soil nutrient status and the mechanism of soil organo-mineral carbon stabilization needs to be re-evaluated, because larger SOC stocks were found in the mineral than in the humus soil horizon. We suggest evaluating enhanced microbial transformation of soil organic matter and the mycorrhizal organic nutrient uptake in relation to larger plant biomass/litter production in nutrient rich sites resulting to higher SOC stock accumulation in deeper soil layers. In addition for the organo-mineral carbon stabilization, we also suggest further model development accounting for the soil nutrient status through evaluating the effect of topography on sorting of the parent material, and its silt and clay complexes.

Our study is very useful for developing accurate soil carbon and Earth system models. Furthermore, developing accurate models that would account for the soil nutrient status as one of the key controls affecting the soil organic matter production and SOC stabilization improves estimation of feedback of global warming on SOC stock temperature sensitivity and soil $CO_2$ efflux, national reporting of soil carbon stock changes for UNFCCC, and implications of decisions mitigating the climate change effects on soil carbon stocks.

## Appendix A:  Models of fraction of absorbed radiation for observed and equilibrium forest

The fraction of photosynthetically active absorbed radiation ($f_{APAR}$) for observed state forest was calculated based on measurements of Swedish forest inventory as in Härkönen et al. (2010). For the main tree species $f_{APAR}$ was also well correlated with the stand basal area ($r^2$ was 0.85, 0.86, and 0.88 for pine, spruce, and deciduous stands respectively, coefficients of regressions in Table A1). The observed state forest $f_{APAR}$ varied between 0 and maximum close to 1 (Fig. A1).

The equilibrium forest $f_{APAR}$ values were assumed to be in range between the median and the maximum fraction of observed state forest $f_{APAR}$ for given species, latitudinal degree, and site productivity class (indicated by the height of largest tress at 100 years of stands age). The equilibrium forest $f_{APAR}$ values were set to 70th percentile of maximum ($f_{APAR70}$) for given species, latitudinal degree, and site productivity class. We selected 70th percentile out of range from 50th to 95th, because the modelled soil carbon distributions with the litter input from biomass of $f_{APAR70}$ best agreed with measured soil carbon distributions (Fig. S2). The $f_{APAR70}$ values specific for pine, spruce, and deciduous stands were first modelled by regression models with latitude ($f_{APAR70LAT}$) (Table A2) and then reduced by the difference between the modelled $f_{APAR70}$ by regression models with site productivity index (H100) ($f_{APAR70H100}$) (Table A1) and maximum $f_{APAR70H100}$ ($f_{APAR70} = f_{APAR70LAT} + f_{APAR70H100}$ - maximum $f_{APAR70H100}$). The $f_{APAR70}$ values equaled the $f_{APAR70LAT}$ values only for the maximum productivity class, otherwise it was reduced.

## Appendix B:  Models of forest dry weight biomass (kg ha$^{-1}$) with $f_{APAR}$.

We fitted species specific exponential regression models between the biomass components (stem, branch, foliage, stump, coarse-roots, fine-roots) of observed state forest and the observed fraction of absorbed radiation ($f_{APAR}$) (scatistics of the regression models in Table B1). The biomass components derived with allometric models (measured) and those derived with $f_{APAR}$ models (modeled) showed strong correlations (Fig. B1). In order to model the longterm mean forest biomass "equilibrium forest biomass" we applied the $f_{APAR}$ biomass models to the modeled $f_{APAR70}$ values.

## Appendix C:  Models of understory vegetation.

We used Swedish forest inventory ground vegetation coverage (%) data visually monitored between 1993 and 2002 on 2440 plots around Sweden with altogether 4472 observations separately for species of forest floor vegetation /or their classes (Table S3). In order to derive the ground vegetation biomass and to apply the coverage/biomass conversion functions (Lehtonen et al., manuscript), we grouped the species coverage observations into five functional types (dwarf-shrubs, herbs, grasses, moss, and lichen) (Table S3). The applied coverage/biomass conversion functions estimated sepa-

rately the above- and below-ground biomass components for dwarf-shrubs, herbs, and grasses, and total biomass for moss, and lichen.

Except the understory coverage, the forest inventory data also contained basic tree dimensions (diameter and height of trees) and stand variables (species dominance, age, basal area, site productivity class indicated by the height of largest tress at 100 years of stands age), and also we linked the plots by their closest proximity to SMHI weather stations with weather data (air temperature, precipitation) and location attributes of the weather stations (latitude, longitude, altitude).

We built linear models for dry weight biomass of understory vegetation ($kg\,ha^{-1}$) in a two level selection of the predictors from stand, weather and location variables. First, we selected the predictors into linear models by using R package "Mass" and its stepwise model selection by exact AIC (Venables and Ripley, 2002). Second, we refined the model by using "relaimpo" R package estimating usefulness (Grömping, 2006), or relative importance for each of the predictors in the model, and by selecting only predictors with relative importance $\geq 0.1$. The general form of the models was:

$$y_i = a + b_1 x_1 + \ldots + b_n x_n + \varepsilon, \tag{C1}$$

Where $y_i$ is the understory dry weight biomass ($kg\,ha^{-1}$), $x_1 \ldots x_n$ are the predictors, $a, b_1 \ldots b_n$ are parameters of the $i^{th}$ understory functional type (Table C1), and $\varepsilon$ is the residual error. Statistics of the models are shown in Table C1. Scatter plots between the measured coverage derived biomass and modelled dry weight biomass ($kg\,ha^{-1}$) of the functional types of ground vegetation for the forests in their observed state close to the estimated equilibrium are shown on Fig. S9.

### Code and data availability

The source codes of the Yasso07, Q and CENTURY models used in this paper are available through the supplementary material. Data used in this study can be available directly by contacting the authors.

*Acknowledgements.* We thank the Finnish Ministry of Environment and the Finnish Ministry of Agriculture and Forestry for funding this work through the Metla project 7509 'Improving soil carbon estimation of greenhouse gas inventory', and Academy of Finland for funding the mobility projects 276300 and 276602. We would like to thank the editor and the reviewers for their valuable comments improving the manuscript.

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

**Table 1.** Description of the Swedish Forest Soil Inventory (SFSI) data reduction of soil sorting of parent material and humus types; SFSI conversion estimate of soil classes of soil moisture to numerical representation of soil water content; and SFSI conversion estimate of classes to numerical representation of soil texture (sand, silt, and clay content for sediments by Lindén (2002) and for tills by Albert Atterberg's distribution of the different grain size fractions).

| SORTING PARENT MATERIAL | | HUMUS TYPE | | MOISTURE | | |
| --- | --- | --- | --- | --- | --- | --- |
| SFSI | REDUCED | SFSI | REDUCED | SFSI | SFSI | NUMERIC |
| Bedrock | Bedrock | Moder | No-peat | | Water | Long-term |
| Poorly sorted sediments | Unsorted | Mor 1 | No-peat | | level (m) | moisture % |
| Tills | Unsorted | Mor 2 | No-peat | Dry | <2 | 10 |
| Well sorted sediments | Sorted | Mull | No-peat | Fresh | 1-2 | 20 |
| | | Mull-Moder | Peat | Fresh-moist | <1 | 30 |
| | | Peat | Peat | Moist | <0.5 | 50 |
| | | Peat-Mor | Peat | | | |

| TEXTURE | | | | | | |
| --- | --- | --- | --- | --- | --- | --- |
| SFSI | NUMERIC | | | | | |
| | SEDIMENTS | | | TILLS | | |
| | Sand % | Silt % | Clay % | Sand % | Silt % | Clay % |
| Bedrock | 0 | 0 | 0 | 0 | 0 | 0 |
| Boulder | 0 | 0 | 0 | 0 | 0 | 0 |
| Gravel | 10 | 0 | 0 | 10 | 0 | 0 |
| Coarse-sand | 40 | 5 | 0 | 40 | 5 | 0 |
| Sand | 80 | 10 | 0 | 45 | 10 | 0 |
| Fine-sand | 70 | 25 | 5 | 55 | 15 | 0 |
| Coarse-silt | 50 | 40 | 10 | 65 | 20 | 5 |
| Fine-silt | 10 | 75 | 15 | 55 | 35 | 10 |
| Clay | 0 | 65 | 35 | 0 | 85 | 15 |
| Peat | 0 | 0 | 0 | 0 | 0 | 0 |

**Table 2.** Descriptive characteristics (mean, confidence interval, $1^{st}$, $50^{th}$, and $99^{th}$ percentile) of selected variables (n = 3230 samples). The values of the bulk density, cation exchange capacity, base saturation, C/N ratio, and pH are shown only for BC soil horizon (fixed 45–50 cm depth from the ground surface) due to the strong correlation to the total soil carbon stock. The soil was cut off at 1 meter. The productivity class (H100, m) is an approximation of the site fertility expressed as the height of trees at 100 years of age. Stand and understory biomass, and litter input are modelled values for approximated equilibrium conditions based on observed state measurements.

| | Mean | CI | $1^{st}$ percentile | $50^{th}$ percentile | $99^{th}$ percentile |
|---|---|---|---|---|---|
| Total soil carbon stock ($tC\,ha^{-1}$) | 93.24 | 1.95 | 17.02 | 79.68 | 308.68 |
| Humus carbon stock ($tC\,ha^{-1}$) | 33.29 | 1.17 | 3.89 | 22.82 | 176.66 |
| Mineral soil carbon stock ($tC\,ha^{-1}$) | 66.82 | 1.7 | 6.92 | 54.81 | 273.91 |
| Depth of humus (cm) | 10.52 | 0.27 | 1 | 8 | 36 |
| Depth of soil (cm) | 93.37 | 0.6 | 18 | 99 | 99 |
| Stoniness (%) | 39.91 | 0.54 | 3.96 | 42.37 | 65.05 |
| Bulk density of BC ($g\,dm^{-3}$) | 1267.1 | 5.5 | 790.55 | 1294.9 | 1522.13 |
| Cation exchange capacity of BC ($mmol_c\,kg^{-1}$) | 23.94 | 1.28 | 1.53 | 12.33 | 203.25 |
| Base saturation of BC (%) | 36.44 | 1.02 | 4.33 | 25.73 | 100 |
| C/N ratio of BC | 16.5 | 0.35 | 3.33 | 14.98 | 62.45 |
| pH of BC | 5.17 | 0.02 | 4.36 | 5.08 | 7.26 |
| Silt content (%) | 19.98 | 0.57 | 0 | 15 | 85 |
| Clay content (%) | 3.16 | 0.25 | 0 | 0 | 35 |
| Sand content (%) | 51.25 | 0.63 | 0 | 55 | 80 |
| Long-term soil moisture (%) | 22.36 | 0.2 | 10 | 20 | 30 |
| Mean air temperature (°C) | 4.63 | 0.09 | -0.44 | 5.34 | 8.47 |
| Total precipitation (mm) | 697.87 | 7.13 | 392.54 | 637.11 | 1154.55 |
| Nitrogen deposition ($kgN\,ha^{-1}\,y^{-1}$) | 7.17 | 0.14 | 2.35 | 6.56 | 17.67 |
| Productivity class (H100, m) | 23.61 | 0.21 | 12 | 23 | 36 |
| Total stand biomass ($tC\,ha^{-1}$) | 56.02 | 1.39 | 1.34 | 51.14 | 156.52 |
| Total understory biomass ($tC\,ha^{-1}$) | 2.69 | 0.05 | 0.96 | 2.37 | 6.02 |
| Total litterfall input ($tC\,ha^{-1}$) | 3.17 | 0.03 | 1.65 | 3.07 | 5.28 |


**Table 3.** Description of models and data inputs relevant for this study.

| Model | Yasso07 | Q | CENTURY v. 4.0 soil submodel |
|---|---|---|---|
| Time step | Year | Year | Month |
| Parametrization | Global | Scandinavian | Combined global with site specific |
| Carbon pools | Labile (acid -, water -, and ethanol-soluble and non-soluble), recalcitrant (humus) | Cohorts (foliage, stems, branches, coarse roots, fine roots, "grass"), soil organic | Litter (surface structural and metabolic, belowground str. and met.), surface microbial, soil organic matter (active, slow and passive) |
| Biomass | Biomass components estimated by allometric biomass functions and provided stand data for litter input estimation | | |
| Litter amount | Annual or monthly fractions of biomass components (species specific, same total litter inputs for all models) | | |
| Litter quality | Litterature based solubilities | Estimated cohorts qualities | C/N ratios and lignin/N ratios |
| Temperature air | Annual mean, monthly amplitude | Annual mean | Max and min monthly mean |
| Precipitation | Annual total | – | Monthly total |
| Soil properties | – | – | Bulk density, sand, silt, and clay content |
| Soil depth (m) | 1 | – | 0.2 |

**Table A1.** Parameter estimates and their standard errors of the $f_{APAR}$ regressions with the stand basal area (BA, $\mathrm{m^2\,ha^{-1}}$), and the $f_{APAR70LAT}$ and $f_{APAR70H100}$ regressions with the latitude (LAT, °) and with the productivity class (H100, m) for Scots pine, Norway spruce, and deciduous stands.

| $f_{APAR} = a*BA/(b+BA)$ | a±SE | b±SE | c±SE | $adj.R^2$ |
|---|---|---|---|---|
| pine | 1.00±0.03 | 11.75±0.81 | | 0.85 |
| spruce | 1.17±0.03 | 10.67±0.87 | | 0.86 |
| deciduous | 1.13±0.06 | 7.41±1.15 | | 0.88 |
| $f_{APAR70LAT} = LAT/(a+b*LAT)+c$ | | | | |
| pine | -9976.00±3691.00[a] | 143.00±54.16[b] | 0.72±0.02 | 0.92 |
| spruce | -2689.00±3507.00[c] | 35.33±50.25[d] | 0.97±0.09 | 0.74 |
| $f_{APAR70LAT} = a+b*LAT$ | | | | |
| deciduous | 1.36±0.28 | -0.01±0.01[e] | | 0.26 |
| $f_{APAR70H100} = a*e^{(b/H100)}$ | | | | |
| pine | 0.86±0.02 | -5.22±0.41 | | 0.89 |
| spruce | 0.97±0.01 | -2.85±0.22 | | 0.86 |
| deciduous | 0.94±0.02 | -2.63±0.50 | | 0.51 |

$p < 0.001$ for all parameters except for [a] 0.023, [b] 0.024, [c] 0.461, [d] 0.498, and [e] 0.076.


**Table B1.** Parameter estimates and their standard errors for the coefficients of the dry weight biomass $(\mathrm{kg\,ha^{-1}})$ models with the fraction of absorbed radiation $(y = ab^{fAPAR})$ for Scots pine, Norway spruce, and deciduous stands.

| $y = ab^{fAPAR}$ | species | a±SE | b±SE | $adj.R^2$ |
|---|---|---|---|---|
| branch | pine | 610.23±21.04 | 121.59±5.97 | 0.92 |
| | spruce | 877.27±34.54 | 54.16±2.46 | 0.92 |
| | deciduous | 289.72±26.46 | 155.51±15.84 | 0.89 |
| fineroot | pine | 422.03±12.68 | 20.51±0.91 | 0.84 |
| | spruce | 316.68±13.82 | 15.19±0.78 | 0.80 |
| | deciduous | 452.63±27.72 | 14.50±1.03 | 0.82 |
| foliage | pine | 361.43±24.10 | 86.09±8.22 | 0.71 |
| | spruce | 766.32±40.28 | 33.32±2.03 | 0.83 |
| | deciduous | 141.11±28.35 | 70.63±15.99 | 0.56 |
| root | pine | 703.16±26.17 | 183.00±9.62 | 0.92 |
| | spruce | 628.69±32.37 | 113.44±6.67 | 0.90 |
| | deciduous | 358.64±33.27 | 149.85±15.51 | 0.89 |
| stem and bark | pine | 1793.22±83.82 | 253.68±16.66 | 0.89 |
| | spruce | 974.03±72.35 | 229.02±19.26 | 0.86 |
| | deciduous | 971.59±97.63 | 160.86±18.02 | 0.88 |
| stump | pine | 231.70±10.27 | 214.43±13.39 | 0.89 |
| | spruce | 170.77±10.33 | 129.22±8.91 | 0.88 |
| | deciduous | 79.78±8.39 | 215.51±25.17 | 0.87 |

$p < 0.001$ for all parameters.

**Table C1.** Parameter estimates and their standard errors for the coefficients of the forest understory vegetation dry weight biomass ($\mathrm{kg\,ha^{-1}}$) models (Eq. C1) for functional types (1-dwarfshrubs, 2-herbs, 3-grasses, 4-mosses and 5-lichens) with intercept (a) and n number of predictors (b1- age (years), b2 – basal area ($\mathrm{m^2\,ha^{-1}}$), b3 – annual air temperature ($^\circ$C), b4 - latitude ($^\circ$), b5 – H100 (height of trees at 100 years of age, m), b6 – H100 of spruce trees (m), b7 – H100 of pine trees (m), b8- pine dominance (0/1), b9-spruce dominance (0/1)). For the latin names of species included into understory functional types see Table S3.

| W | | a±SE | b1±SE | b2±SE | b3±SE | b4±SE | b5±SE | b6±SE | b7±SE | b8±SE | b9±SE | $adj.R^2$ |
|---|---|---|---|---|---|---|---|---|---|---|---|---|
| Above-ground | 1 | 24.28±0.32 | 0.13±0.01 | -0.43±0.02 | | | | | | 7.13±0.33 | | 0.29 |
| | 2 | -82.13±6.8 | | | -0.1±0.1[a] | 1.23±0.1 | | 0.77±0.03 | | | | 0.12 |
| | 3 | 4.07±0.30 | | -0.16±0.01 | | | | 0.27±0.01 | | -1.36±0.15 | | 0.21 |
| | 4 | 32.9±0.62 | | | | | -0.78±0.04 | | 0.48±0.06 | 3.66±0.3 | 5.76±0.29 | 0.22 |
| | 5 | 19.91±0.57 | | -0.13±0.01 | | | | -0.45±0.02 | | 6.31±0.29 | | 0.25 |
| | total | 43.68±0.29 | 0.12±0.01 | -0.41±0.01 | | | | | | 6.34±0.3 | | 0.30 |
| Below-ground | 1 | -256.3±3.5 | 0.1±0.01 | -0.35±0.02 | | 5.05±0.06 | | | | 8.56±0.35 | | 0.75 |
| | 2 | -89.34±7.85 | | | -0.03±0.1[b] | 1.4±0.12 | | 0.78±0.04 | | -4.97±0.27 | | 0.19 |
| | 3 | 5.97±0.37 | | -0.19±0.01 | | | | 0.32±0.01 | | -1.78±0.19 | | 0.21 |
| | total | -251.9±3.3 | | -0.2±0.01 | | 5.15±0.05 | | | | | | 0.7 |
| Total | | -222.7±4.0 | 0.12±0.01 | -0.44±0.02 | | 4.9±0.07 | | | | | | 0.67 |

$p < 0.001$ for all parameters except for [a] $p = 0.44$, and [b] $p = 0.84$.

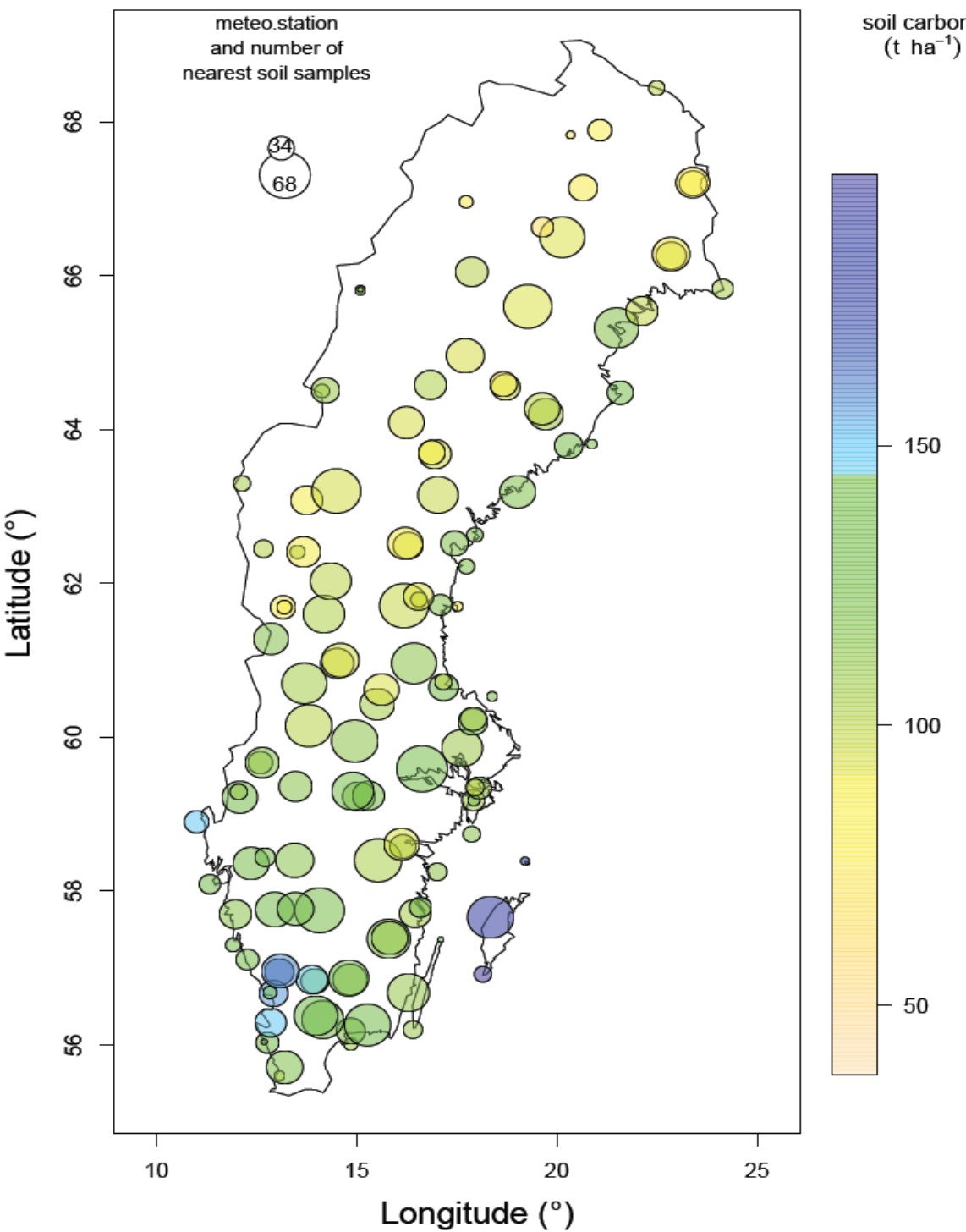

**Figure 1.** Geographical locations of meteorological stations with corresponding number of nearest soil samples (n, size of the circle) and their mean measured soil organic carbon stock ($tC\,ha^{-1}$, color of the circle) across Sweden.

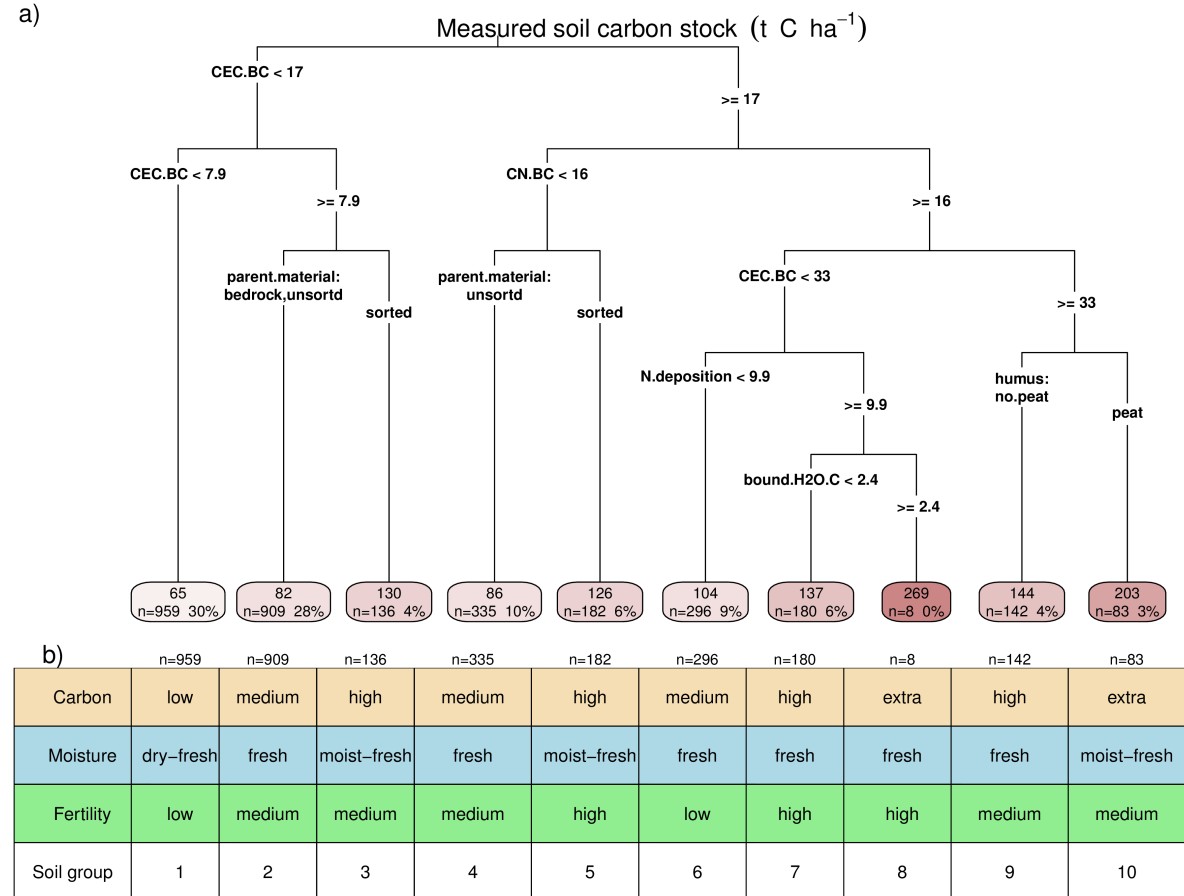

**Figure 2.** a) Classification/regression tree for the measured soil carbon stock ($\mathrm{t\,C\,ha^{-1}}$), soil physicochemical properties and site environmental characteristics; the cation exchange capacity of BC horizon (CEC.BC, ($\mathrm{mmol}_c\,\mathrm{kg^{-1}}$)), the C/N ratio (CN.BC), the nitrogen deposition (N.deposition $\mathrm{kgN\,ha^{-1}\,y^{-1}}$), the highly bound soil water of C horizon (bound.H2O.C, %), and soil class variables as type of sorted or unsorted soil parent material and humus type. Note that variables used to calculate the soil carbon stock (bulk density, carbon content, depth, and stoniness) were excluded from the regression tree analysis. The values in the leaves of the tree show for the distinct environmental conditions mean soil carbon stock ($\mathrm{t\,C\,ha^{-1}}$), number and percentage of samples. b) The interpretation of 10 physicochemical soil groups of the regression tree model into the levels of carbon, soil moisture, and fertility roughly increasing from left to right.

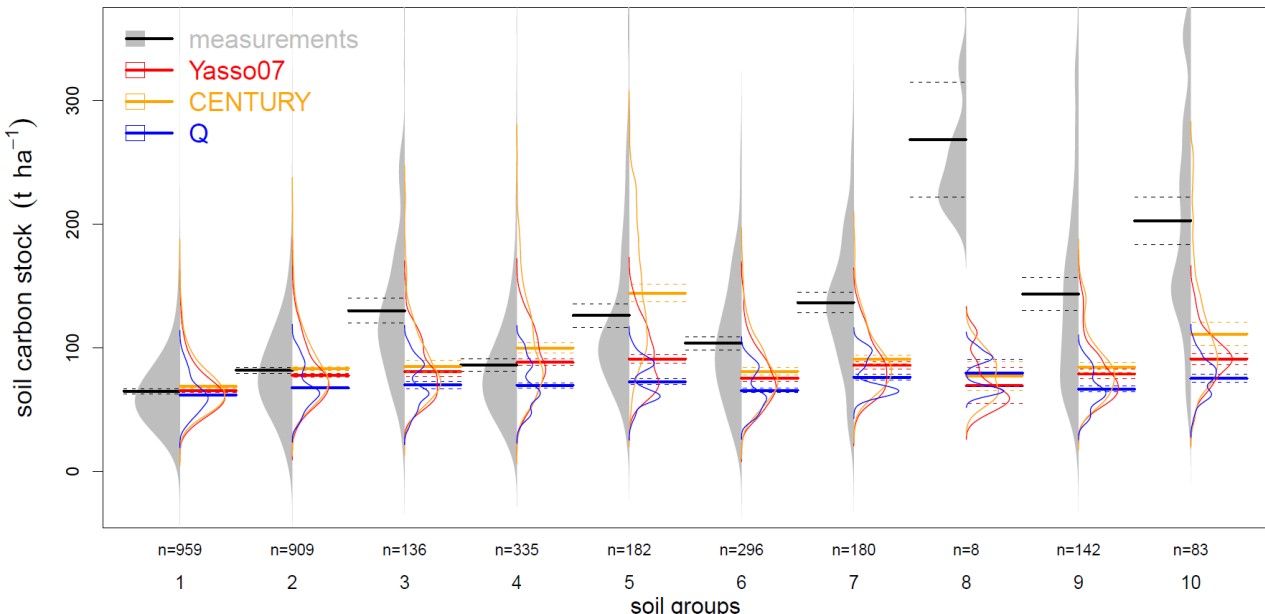

**Figure 3.** Bean plot of density functions for 10 physicochemical groups of the soil carbon $(tC\,ha^{-1})$ measurements (grey fill) and estimates simulated by the soil carbon models Yasso07, CENTURY, and Q with the litter input derived from the equilibrium forest. The thin lines are the density distributions. The thick lines are the group means and dashed lines are their confidence intervals. The n is number of samples. For description of group levels of SOC stocks, moisture, and fertility see Fig.2 and Table S1.

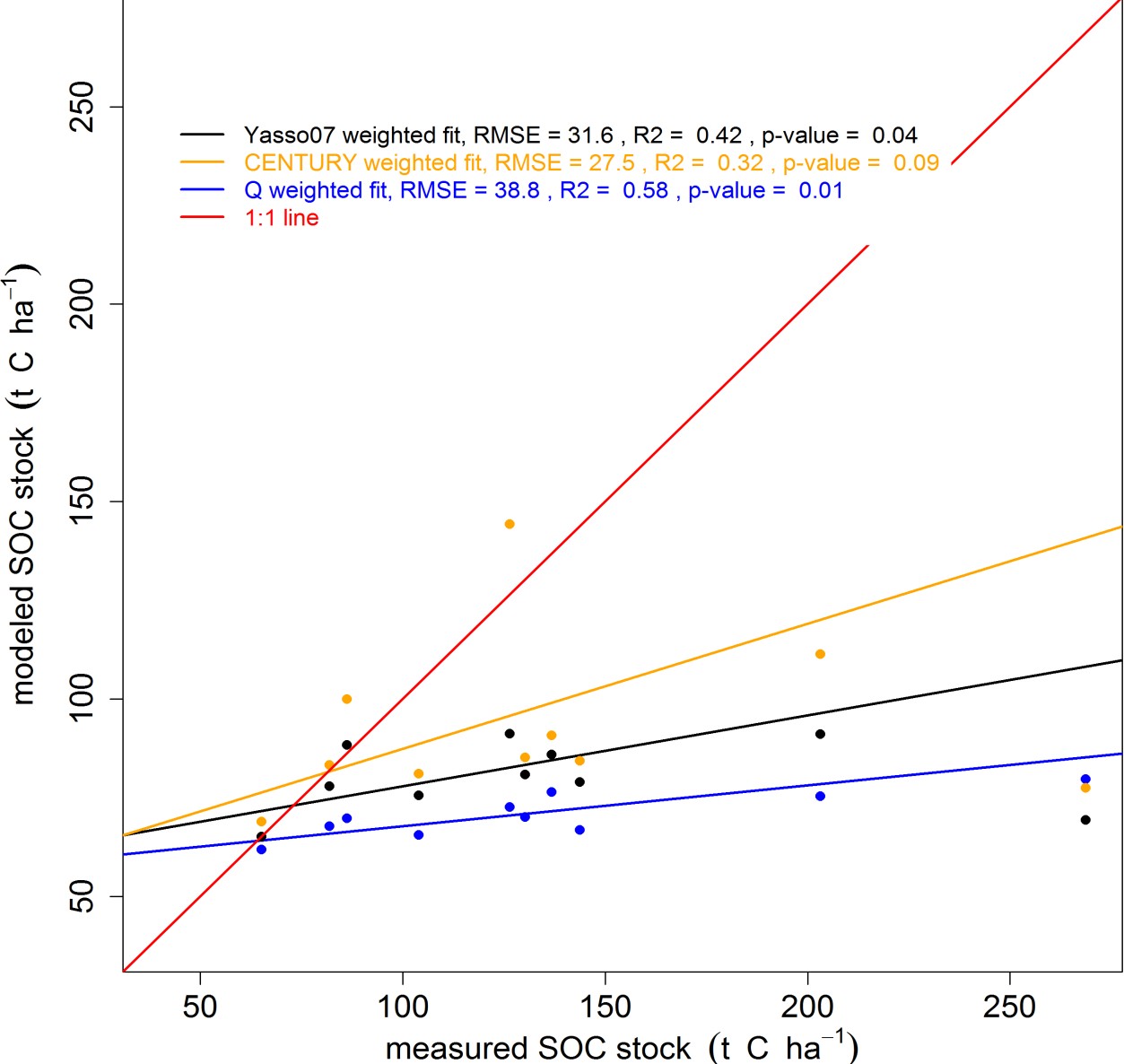

**Figure 4.** Scatter plot between mean measured and mean modeled soil organic carbon stocks $(\mathrm{t\,C\,ha^{-1}})$ for 10 physicochemical groups for Yasso07, CENTURY and Q models. Data were fitted with weighted linear regression (lines). The number of samples in each group was used as weights for fitting and also as weights for the weighted mean of squared differences between the modeled and measured values (MSE, $\mathrm{t\,C\,ha^{-1}}$). The RMSE is the square root of MSE. The $r^2$ is the proportion of explained variance. The p-value is the calculated probability that the fit is significant.

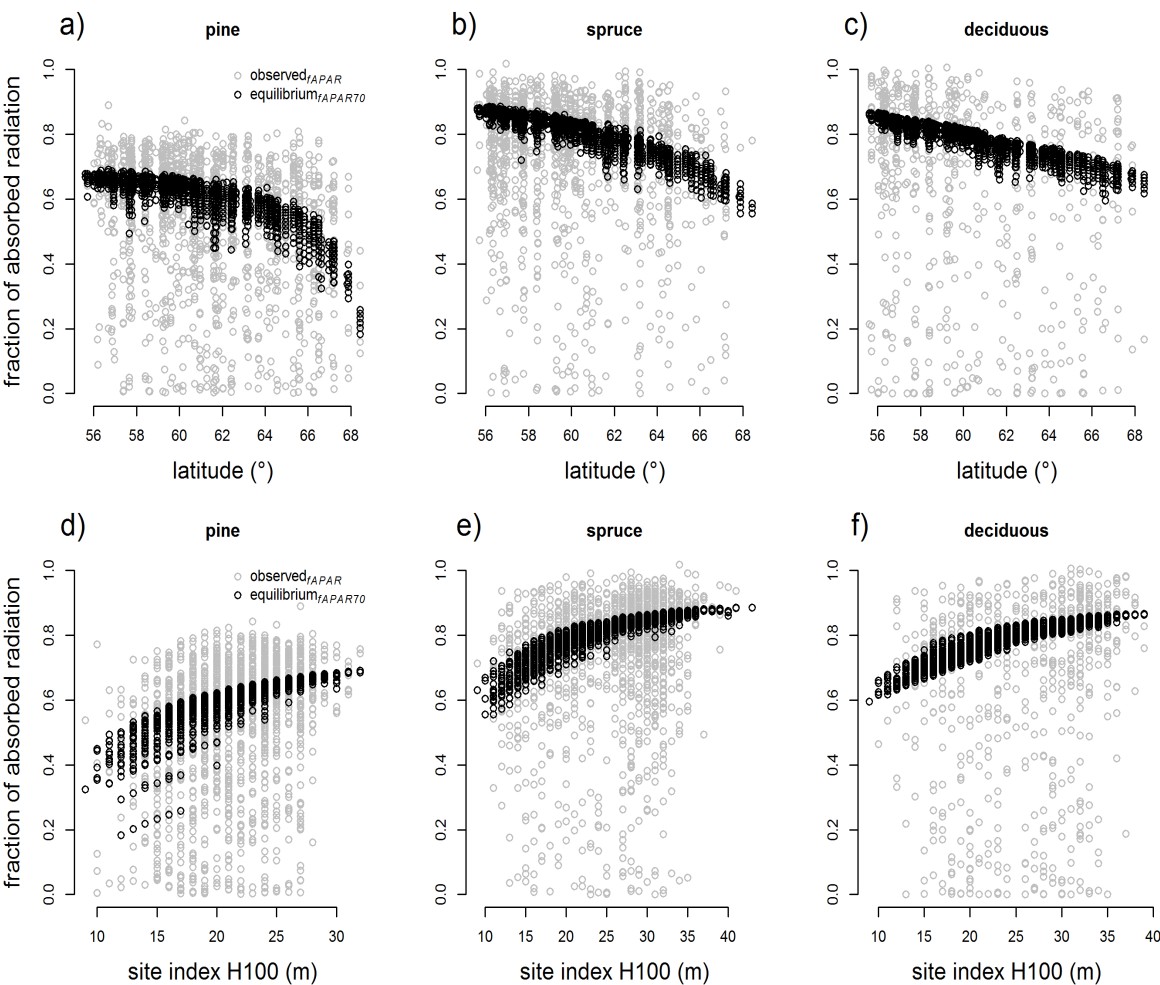

**Figure A1.** Observed fraction of absorbed radiation ($f_{APAR}$, estimated as in Härkönen et al., 2010) (observed $f_{APAR}$) and equilibrium $f_{APAR}$ (modeled $f_{APAR70}$) which was set to 70th percentile of maximum $f_{APAR}$ for given species, latitudinal degree, and site productivity class. Panels a), b), and c) show relation between $f_{APAR}$ and latitude (°) for forest stands dominant by Scots pine, Norway spruce and deciduous species, whereas panels d), e), and f) show relation between $f_{APAR}$ and site index H100 (height of dominant trees at 100 years in meters).

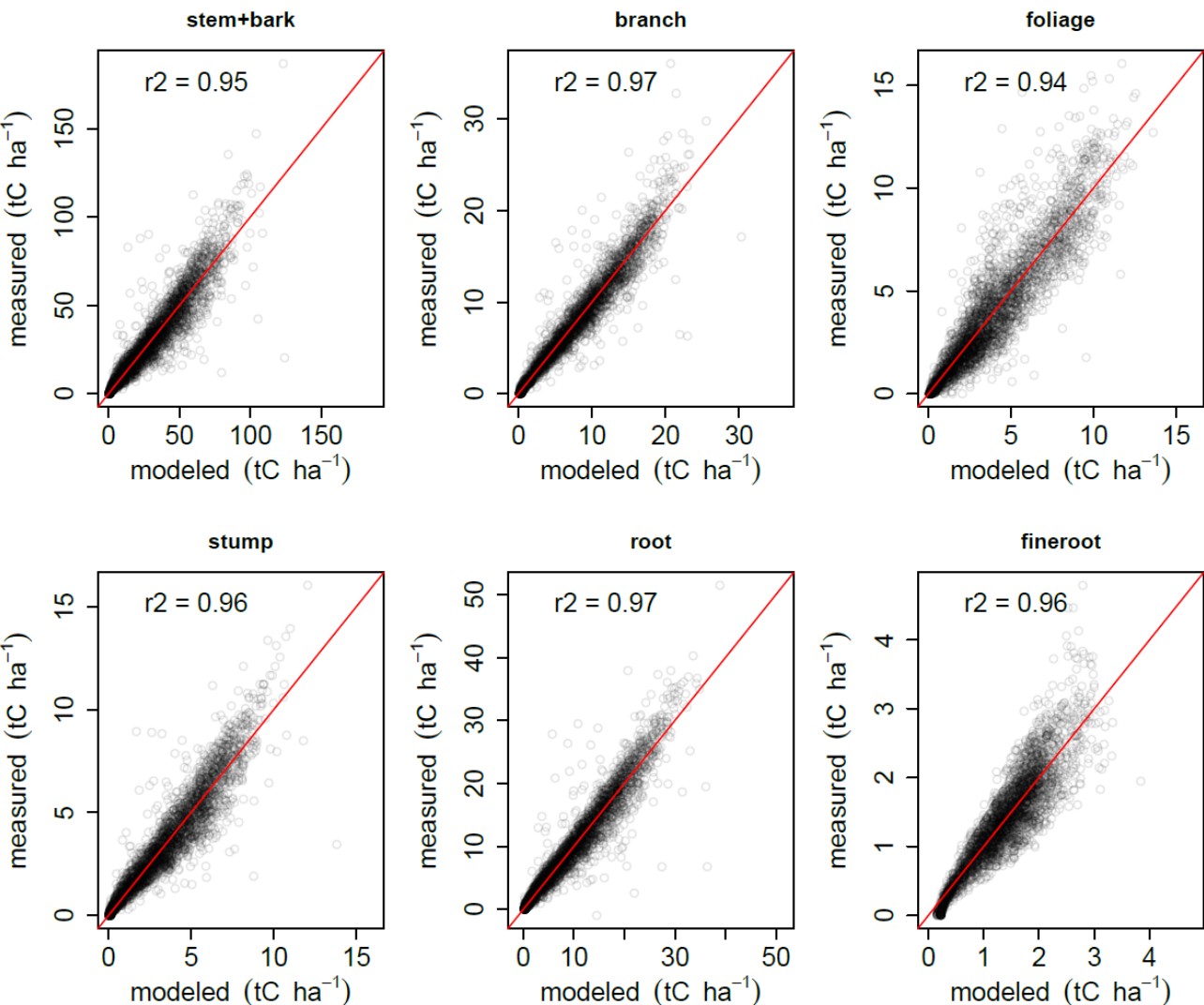

**Figure B1.** Scatter plots (n = 3698 in all panels) for the dry weight tree biomass components $(\mathrm{tC\,ha^{-1}})$ between "modelled" (estimated based on fraction of absorbed radiation, $f_{APAR}$, and our $f_{APAR}$ models) and "measured" (estimated based on basic tree stand dimensions and allometric biomass models). The $r^2$ values represent the coefficient of determination indicating how close the modeled values fit the measured values.