# Peer review of "Underestimation of boreal soil carbon stocks by mathematical soil carbon models linked to soil nutrient status"

_Biogeosciences, 2015_

## Short Comment (SC1) · 4 Feb 2016

General comments This is an interesting paper. Three structurally quite different soil carbon models give very similar predictions of forest soil carbon stocks when they are driven by the same litter inputs and differ also similarly from observations. The critical question is why they fail in their predictions for 22% of the test sites. The authors attribute the failure to weaknesses in how the models handle soil nutrient status. This might well be the case, but such a failure can come from two quite different sources. On one hand, is the litter input correctly calculated? The procedure used to generate litter input is not transparent. The calculation is based on fAPAR (the fraction of absorbed photosynthetically active radiation) but the maximum/potential value of absorbed radi-

ation seems to be ignored. However, both the potential production and fAPAR vary with the nutrient status of the stand. In the end, it seems to me that the procedure generates tree biomasses and thus litter production only depending on latitude; this will ignore the large regional differences in nitrogen deposition that play an important role in tree productivity, likely leading to an underestimate of litter production in high deposition areas. On the other hand, it is clear that soil nitrogen modifies the carbon use efficiency of decomposers; increasing nitrogen availability increases CUE, which increases soil carbon stocks (Ågren et al. 2001, Franklin, et al. 2003). In all three models, inclusion of either of these two factors would improve the model performance at the high nutrient sites.

Specific comments 1. Line 78. effects should be affects 2. Line 221. It is not clear what is meant by "the 2012Q model". Should it be 2011 or 2013? 3. Line 343. Why should decreased microbial demand for nitrogen lead to increased soil carbon? 4. Line 387. Why should inorganic nutrient uptake by mycorrhiza lead to underestimated SOC stocks on medium-highly productive sites? Cited literature Franklin, O., et al. (2003). "Pine forest floor carbon accumulation in response to N and PK additions - Bomb 14C modelling and respiration studies." Ecosystems 6: 644-658.

Ågren, G. I., et al. (2001). "Combining theory and experiment to understand effects of inorganic nitrogen on litter decomposition." Oecologia (Heidelb.) 128: 94-98.

---

## Referee Comment (RC1) · Dr Golubyatnikov (Referee) · 9 Feb 2016

Authors evaluated soil organic carbon stock for Swedish forest using models Yasso07, Q, CENTURY and compared the model results with the Swedish forest soil inventory data. They described the obtained results very accurate and comprehensively.

Remarks: 1. Is phrase "i.e. samples with SOC stock below 0.01 and 99.9 percentile" (line 103) correct? 2. It's not necessary to reintroduce the abbreviations (for example, line 102). 3. Units for turnover rate are necessary (lines 159-164). 4. Section 2.2 duplicates the information from lines 64-80.

Authors used linear functions for biomass of vegetation types. According to Tabl.C1

all(!!!) functions for aboveground biomass have R2<0.5 and only one function for be-lowground biomass has R2>0.5. Therefore, these functions do not reflect the realistic interdependences and increase the model mistakes.

It is not clear what authors wanted to show by this manuscript. From the presented results it follows that models of some processes do not accurately reflect these real processes. But it is evident and not new! Another conclusion of the article is also obvious: data for model essentially impact the model results.

I think this manuscript can not be published

---

## Author Comment (AC1) · 16 Feb 2016

Author's reply to Prof G. I. Ågren (Referee):

!Referee: referee's comments #Authors: author's reply

GENERAL COMMENTS

!Referee: This is an interesting paper.

**Authors: Thank you, we appreciate all your comments, considered them carefully, and reply below to each of them! In addition the PDF version of our reply and the marked up manuscript with highlighted changes is provided in the supplement of our comment.**
[Figure]

!Referee:Three structurally quite different soil carbon models give very similar predictions of forest soil carbon stocks when they are driven by the same litter inputs and differ also similarly from observations. The critical question is why they fail in their predictions for 22% of the test sites. The authors attribute the failure to weaknesses in how the models handle soil nutrient status. This might well be the case, but such a failure can come from two quite different sources. On one hand, is the litter input correctly calculated?

**Authors: Yes the litter input is calculated correctly, as we are aware that the correct calculation of the litter input is essential for the simulation of the soil carbon sequestration and the estimation method has large influence on the sequestered soil carbon. E.g. see SOC and litter relations in supplement figure FS6 and results lines 306 - 310.**

!Referee:The procedure used to generate litter input is not transparent.

**Authors: We are aware that our description of the novel approach of litter input estimation may not be transparent in general concept in Sect. 2.1.1 "Biomass and litterfall estimates", therefore we added detailed descriptions for reproducing the methods to appendices (Appendices A, B, and C, Tables A1, B1, and C1, and Figures A1, B1, and S9). At first, the novel method could seem complicated compared to the estimation by using only the allometric biomass models. However, the measurements of actual state forest could not be applied directly to biomass models in order to derive the long-term litter inputs due to differences in stand age classes and our method to remove the effect of the actual stand development was crucial for estimating long-term mean litter input correctly.**

!Referee:The calculation is based on fAPAR (the fraction of absorbed photosynthetically active radiation) but the maximum/potential value of absorbed radiation seems to be ignored. However, both the potential production and fAPAR vary with the nutrient status of the stand. In the end, it seems to me that the procedure generates tree biomasses and thus litter production only depending on latitude;

[Figure]

**Authors: We are sorry that you partly misunderstood whether the maximum/potential value of absorbed radiation was taken into account. What we meant to describe was that fAPAR was based on the field data, the maximum observed fAPAR was certainly taken into account, and it was specific for latitude and nutrient status, and served as a prerequisite for the estimated 70th percentile of fAPAR (fAPAR70). The nutrient status was in our data represented by a productivity class (H100, height of the dominant trees at the age of 100 years in meters). Both latitude and the H100 data were used in estimation of the fAPAR70 values (Appendix A1 lines 508 - 513, Table A1 and Fig. A1). We think that adding panels showing the relation between modeled fAPAR70 and H100 data into Fig. A1 will clear the confusion about relation between fAPAR and site productivity/nutrient status (see attached updated Fig. A1).**

!Referee: this will ignore the large regional differences in nitrogen deposition that play an important role in tree productivity, likely leading to an underestimate of litter production in high deposition areas.

**Authors: Figure 2 in this reply shows that productivity class (H100) of deciduous, pine, and spruce forests used in this study for the long-term litter input modelling was well correlated with Nitrogen deposition data (panels a, b, and c). However if using the actual state forests measurements directly, with only the allometric biomass models approach, the forest stage development masked the relationship between the nutrient status and the litterfall estimates (actual state forest litter in panels d, e, and f). In our approach with the stage development set to a 70th percentile of the maximum production potential, the litterfall estimates (long-term mean litter) reflected well the differences in Nitrogen deposition (panels g, h, and i).**

!Referee: On the other hand, it is clear that soil nitrogen modifies the carbon use efficiency of decomposers; increasing nitrogen availability increases CUE, which increases soil carbon stocks (Ågren et al. 2001, Franklin, et al. 2003). In all three models, inclusion of either of these two factors would improve the model performance at the high nutrient sites.

**Authors: We added your comment into discussion, by reformulating sentence on lines 343-345, complementing on studies of Fernandez-Martinez et al. 2014, and Manzoni et al. 2012. "Larger net soil carbon accumulation in nutrient rich sites could be attributed to the relative differences in litterfall components (relatively more leaves and branches with higher N content than fine roots), and to higher N availability and carbon use efficiency of decomposers, reduction of respiration per unit of C uptake (Ågren et al. 2001, Manzoni et al. 2012, Fernandez-Martinez et al., 2014)." Manzoni, S., Taylor, P., Richter, A., Porporato, A. and Ågren, G. I.: Environmental and stoichiometric controls on microbial carbon-use efficiency in soils, New Phytol., 196, 79-91, 2012.**

**Authors: We also added citation of Franklin et al. (2003) after the sentence on line 347. "The soils with large N deposition were also highly productive and showed high to exceptionally high SOC stocks (Fig. 2, Fig. 3, soil groups 7 and 8). This was in agreement with fertilization and modelling study of Franklin et al. (2003) showing an increase in soil C accumulation with N addition."**

SPECIFIC COMMENTS

!Referee: 1. Line 78. effects should be affects

**Authors: Effects was changed to affects**

!Referee: 2. Line 221. It is not clear what is meant by "the 2012Q model". Should it be 2011 or 2013? #Authors: We changed it to 2011, because 2011 was the calibration of the model and 2013 was an application on larger regions, no calibration.

!Referee: 3. Line 343. Why should decreased microbial demand for nitrogen lead to increased soil carbon?

**Authors: We reformulated sentence on lines 343-345 as described in general comments**

!Referee: 4. Line 387. Why should inorganic nutrient uptake by mycorrhiza lead to underestimated SOC stocks on medium-highly productive sites?

[Figure]

**Authors: In lines 386-388 based on finding of Orwin et al. (2011) we suggest that not accounting for the available nutrients from the organic (not inorganic) uptake by models contributes to their underestimation of SOC stocks on sites with higher nutrient status. We reformulated the sentence.**

"Expanding on the CENTURY model structure, the MySCaN model incorporating the organic nutrient uptake by mycorrhizal fungi estimated positive effect on SOC accumulation, relatively larger in poor than in fertile sites (Orwin et al.,2011). Therefore, not accounting for the organic nutrient uptake by mycorrhizal fungi by the Yasso07, Q, and CENTURY models probably led to the underestimation of SOC stocks in sites with higher nutrient status."

Orwin, K. H., Kirschbaum, M. U., St John, M. G. and Dickie, I. A.: Organic nutrient uptake by mycorrhizal fungi enhances ecosystem carbon storage: a model-based assessment, Ecol. Lett., 14, 493-502, 2011.

!Referee: Cited literature Franklin, O., et al. (2003)."Pine forest floor carbon accumulation in response to N and PK additions - Bomb 14C modelling and respiration studies." Ecosystems 6: 644-658. Ågren, G. I., et al. (2001). "Combining theory and experiment to understand effects of inorganic nitrogen on litter decomposition." Oecologia (Heidelb.) 128: 94-98.

**Authors: Thank you for providing these references.**

**Authors: Figure captions**

Figure 1. or Figure A1 in our BGD paper. Actual state fraction of absorbed radiation (fAPAR, estimated as in Härkönen et al., 2010) (actual fAPAR) and steady state fAPAR (modeled fAPAR70) which was set to 70th percentile of maximum fAPAR for given species, latitudinal degree, and site productivity class. Panels a), b), and c) show relation between fAPAR and latitude (°) for forest stands dominant by Scots pine, Norway spruce and deciduous species, whereas panels d), e), and f) show relation between

[Figure]

fAPAR and site productivity class (H100, height of dominant trees at 100 years in meters).

Figure 2. Scatterplots between the Nitrogen deposition (kg N ha-1 y-1) and a), b), c) site productivity class (H100, which is the height of the dominant trees at the age of 100 years in meters) , d), e), f) actual state forest litterfall (t C ha-1 y-1), and g), h), i) long-term mean "steady state" forest litterfall (t C ha-1 y-1) for deciduous species, Scots pine, and Norway spruce dominated stands.

Please also note the supplement to this comment:
http://www.biogeosciences-discuss.net/bg-2015-657/bg-2015-657-AC1-
supplement.pdf

―――――――――――――――――――

[Figure]

**BGD**

**Fig. 1.**

[Figure]

a) deciduous

b) pine

c) spruce

d)

e)

f)

g)

h)

i)

Fig. 2.

**Supplement:**

*Boris Ťupek\*, Carina A. Ortiz, Shoji Hashimoto, Johan Stendahl, Jonas Dahlgren, Erik Karltun, and Aleksi Lehtonen*
*\*boris.tupek@luke.fi*

**Referee's comments are highlighted by bold font.**
*# Symbol and font used to indicate author's reply*

General comments
**This is an interesting paper.**
*# Thank you, we appreciate all your comments, considered them carefully, and reply below to each of them!*
*The marked up manuscript with highlighted changes is attached to this file.*

**Three structurally quite different soil carbon models give very similar predictions of forest soil carbon stocks when they are driven by the same litter inputs and differ also similarly from observations. The critical question is why they fail in their predictions for 22% of the test sites. The authors attribute the failure to weaknesses in how the models handle soil nutrient status. This might well be the case, but such a failure can come from two quite different sources. On one hand, is the litter input correctly calculated?**
*# Yes the litter input is calculated correctly, as we are aware that the correct calculation of the litter input is essential for the simulation of the soil carbon sequestration and the estimation method has large influence on the sequestered soil carbon. E.g. see SOC and litter relations in supplement figure FS6 and results lines 306 - 310.*

**The procedure used to generate litter input is not transparent.**
*# We are aware that our description of the novel approach of litter input estimation may not be transparent in general concept in Sect.* 2.1.1 "Biomass and litterfall estimates", *therefore we added detailed descriptions for reproducing the methods to appendices (Appendices A, B, and C, Tables A1, B1, and C1, and Figures A1, B1, and S9). At first, the novel method could seem complicated compared to the estimation by using only the allometric biomass models. However, the measurements of actual state forest could not be applied directly to biomass models in order to derive the long-term litter inputs due to differences in stand age classes and our method to remove the effect of the actual stand development was crucial for estimating long-term mean litter input correctly.*

**The calculation is based on fAPAR (the fraction of absorbed photosynthetically active radiation) but the maximum/potential value of absorbed radiation seems to be ignored. However, both the potential production and fAPAR vary with the nutrient status of the stand. In the end, it seems to me that the procedure generates tree biomasses and thus litter production only depending on latitude;**
*# We are sorry that you partly misunderstood whether the maximum/potential value of absorbed radiation was taken into account. What we meant to describe was that fAPAR was based on the field data, the maximum observed fAPAR was certainly taken into account, and it was specific for latitude and nutrient status, and served as a prerequisite for the estimated 70th percentile of fAPAR ($f_{APAR70}$). The nutrient status was in our data represented by a productivity class (H100, height of the dominant trees at the age of 100 years in meters). Both latitude and the H100 data were used in estimation of the $f_{APAR70}$ values (Appendix A1 lines 508 - 513, Table A1 and Fig. A1). We think that adding panels showing the relation between modeled $f_{APAR70}$ and H100 data into Fig. A1 will clear the confusion about relation between fAPAR and site productivity/nutrient status (see attached updated Fig. A1).*

[Figure]

Figure A1. Actual state fraction of absorbed radiation (f$_{APAR}$, estimated as in Härkönen et al., 2010) (actual f$_{APAR}$) and steady state f$_{APAR}$ (modeled f$_{APAR70}$) which was set to 70th percentile of maximum f$_{APAR}$ for given species, latitudinal degree, and site productivity class. Panels a), b), and c) show relation between f$_{APAR}$ and latitude (°) for forest stands dominant by Scots pine, Norway spruce and deciduous species, whereas panels d), e), and f) show relation between f$_{APAR}$ and site productivity class (H100, height of dominant trees at 100 years in meters).

**this will ignore the large regional differences in nitrogen deposition that play an important role in tree productivity, likely leading to an underestimate of litter production in high deposition areas.**
*#A figure (Fig. R1) in this reply shows that productivity class (H100) of deciduous, pine, and spruce forests used in this study for the long-term litter input modelling was well correlated with Nitrogen deposition data (panels a, b, and c). However if using the actual state forests measurements directly, with only the allometric biomass models approach, the forest stage development masked the relationship between the nutrient status and the litterfall estimates (actual state forest litter in panels d, e, and f). In our approach with the stage development set to a 70$^{th}$ percentile of the maximum production potential, the litterfall estimates (long-term mean litter) reflected well the differences in Nitrogen deposition (panels g, h, and i).*

[Figure]

Figure R1. Scatterplots between the Nitrogen deposition (kg N ha$^{-1}$ y$^{-1}$) and a), b), c) site productivity class (H100, which is the height of the dominant trees at the age of 100 years in meters) , d), e), f) actual state forest litterfall (t C ha$^{-1}$ y$^{-1}$), and g), h), i) long-term mean "steady state" forest litterfall (t C ha$^{-1}$ y$^{-1}$) for deciduous species, Scots pine, and Norway spruce dominated stands.

**On the other hand, it is clear that soil nitrogen modifies the carbon use efficiency of decomposers; increasing nitrogen availability increases CUE, which increases soil carbon stocks (Ågren et al. 2001, Franklin, et al. 2003). In all three models, inclusion of either of these two factors would improve the model performance at the high nutrient sites.**

*#We added your comment into discussion, by reformulating sentence on lines 343-345, complementing on studies of Fernandez-Martinez et al. 2014, and Manzoni et al. 2012.*

"Larger net soil carbon accumulation in nutrient rich sites could be attributed to the relative differences in litterfall components (relatively more leaves and branches with higher N content than fine roots), and to higher N availability and carbon use efficiency of decomposers, reduction of respiration per unit of C uptake (Ågren et al. 2001, Manzoni et al. 2012, Fernandez-Martinez et al., 2014)."

Manzoni, S., Taylor, P., Richter, A., Porporato, A. and Ågren, G. I.: Environmental and stoichiometric controls on microbial carbon-use efficiency in soils, New Phytol., 196, 79-91, 2012.

*#We also added citation of Franklin et al. (2003) after the sentence on line 347.*

"The soils with large N deposition were also highly productive and showed high to exceptionally high SOC stocks (Fig. 2, Fig. 3, soil groups 7 and 8). This was in agreement with fertilization and modelling study of Franklin et al. (2003) showing an increase in soil C accumulation with N addition."

Specific comments
**1. Line 78. effects should be affects**
*#Effects was changed to affects*
**2. Line 221. It is not clear what is meant by "the 2012Q model". Should it be 2011 or 2013?**
*#We changed it to 2011, because 2011 was the calibration of the model and 2013 was an application on larger regions, no calibration.*
**3. Line 343. Why should decreased microbial demand for nitrogen lead to increased soil carbon?**
*#We reformulated sentence on lines 343-345 as described in general comments*
**4. Line 387. Why should inorganic nutrient uptake by mycorrhiza lead to underestimated SOC stocks on medium-highly productive sites?**
*#In lines 386-388 based on finding of Orwin et al. (2011) we suggest that not accounting for the available nutrients from the organic (not inorganic) uptake by models contributes to their underestimation of SOC stocks on sites with higher nutrient status. We reformulated the sentence.*

[revised manuscript text omitted]

---

## Author Comment (AC2) · 25 Feb 2016

**Dr Leonid L. Golubyatnikov's (Referee's) comments are highlighted by bold font.**

*# Symbol and font used to indicate Author's reply.*

**Authors evaluated soil organic carbon stock for Swedish forest using models Yasso07, Q, CENTURY and compared the model results with the Swedish forest soil inventory data. They described the obtained results very accurate and comprehensively.**

*#Thank you for your comments! We appreciate and considered them all, and below we reply to each in detail. Based on your comments we have presented 1 new biomass/*

*litterfall figure and redrawn 2 original biomass modelling figures (attached at the end); and reformulated text by following your remarks and clarifying Sections "2.1.1 Biomass and litterfall estimates" and "5 Conclusions" of our Biogeosciences Discussion (BGD) paper (the marked up version is in the supplement of this comment).*

**Remarks:**

**1. Is phrase "i.e. samples with SOC stock below 0.01 and 99.9 percentile" (line 103) correct?**

*# We reformulated the sentence "SOC stock below 0.01 and above 99.9 percentile"*

**2. It's not necessary to reintroduce the abbreviations (for example, line 102).**

*# We removed "(SOC)" from line 102, "(CEC)" from lines 198, 283, 287,"(SFSI)" from line 321,"(SMHI)" from line 533*

**3. Units for turnover rate are necessary (lines 159-164).**

*# We added short description of TR on the line 153 where it was first introduced* "(TR, the fraction of living biomass that is shed onto the ground per year, unitless)"

**4. Section 2.2 duplicates the information from lines 64-80.**

*# We reformulated section 2.2 by removing information which was previously mentioned in the introduction. The sentence on lines 226-238 was reformulated:* "The Yasso07 model (Tuomi et al., 2009; 2011) is one of the most widely applied SOC models. *The sentence on lines 232-235 was deleted. The sentence on lines 247-248 was reformulated:* "The CENTURY is also one of the most widely applied models."

**Authors used linear functions for biomass of vegetation types. According to Tabl.C1 all (!!!) functions for aboveground biomass have R2<0.5 and only one function for belowground biomass has R2>0.5. Therefore, these functions do not reflect the realistic interdependences and increase the model mistakes.**

[Figure]

*# We are sorry for your possible misunderstanding on the extent in which we used the linear functions for the long-term mean forest biomass and litter input modelling for the soil carbon models. What we meant to describe in Section "2.1.1 Biomass and litterfall estimates" and in Appendices A, B, and C was that we used these linear functions (1) only for the litter input from the understory vegetation, (2) only for the long-term mean conditions "steady state forest", and that (3) the understory vegetation types affected the total understory litterfall with different weights according to their proportion of the total understory litterfall (better models for largely abundant dwarf-shrubs shared most influence than poorer models of scarcer herbs, grasses and lichens).*

*Firstly, it is evident that the forest understory represented the minor part of the total litter input (Fig. 1 of this comment), and that the major part of the litter input originated from the tree stand biomass components which were modeled by the non-linear functions with $R^2$ values close to 0.9 (Fig. 2 of this comment, redrawn Fig. B1, Appendix A and B, Tables A1 and B1). Therefore, when compared to the tree stand whose high model precisions governed the estimated total litter inputs for soil carbon models, and the understory had only small influence on the performances of soil carbon models.*

*Secondly, the variation of observed understory data for the plots close to estimated long-term mean conditions was largely reduced (as juvenile and declining forest phases were excluded) in comparison to the low proportion of explained variance for models presented in Table C1 for forest plots with high variance in understory data due to all stages of forest development. Our application of linear understory models for these plots resulted in much stronger fit between the observed and predicted values (Fig. 3 of this comment as redrawn Fig. S9, mean, min, and max $R^2$ were 0.69, 0.38, and 0.91,respectively).*

*Thirdly, the contribution of understory types to total understory litterfall was largest for the major part of total understory litterfall originating from dwarf-shrubs and mosses*

*(Fig. 1c). Dwarf-shrubs and mosses were predicted for the steady state forest with high $R^2$ values between 0.7 and 0.9 (Fig. 3). The understory vegetation types with the lower $R^2$ values (between 0.38 and 0.66, for herbs, grass, and lichens, Fig. 3) contributed little to total understory litterfall (Fig. 1c). When aiming to evaluate the impact of understory models on performances of SOC models for steady state forests, as in our application, it is correct to consider the larger $R^2$ values of Fig. 3 (especially totals with $R^2$ values close to 0.9, as total understory biomass or litterfall modeled for each functional type separately or in one model highly correlated).Therefore, the influence of these poorer understory models was small on predictions of the understory litter and marginal on predictions of the total forest litterfall and simulated SOC stocks.*

*Note, that SFI observations of forest floor vegetation coverages were not available for 3230 SFI plots with soil data. For the comparison between the understory and the stand biomass based on measurements (Fig. 1), we estimated biomasses for 2440 plots SFI plots which contained the understory data. In order to remove the age class effect on the understory biomass, which was also removed in our BGD paper for plots with soil data by estimating the forest biomass only for steady state, we selected from the 2440 SFI plots only those plots whose estimated fraction of absorbed radiation ($f_{APAR}$, Appendix A) was close to steady state $f_{APAR}$ ($f_{APAR70}$) "steady state forest plots". In order to remove the effect of the actual stand development, which was crucial for estimating long-term mean litter input accurately, we developed functions based on $f_{APAR}$ (Appendices A and B).*

*When regarding the nature of the understory coverage SFI data (visual observations), the lower precision ($R^2$ values below 0.9) of estimated biomasses could be expected even with the most sophisticated ecological models, but the significant p-values of our model parameters with predicted and observed values showing approximately 1:1 relation indicated that the estimates were accurate. Our aim here was to produce accurate biomass/litterfall estimates representing the mean long-term conditions (defined by es-*

*timated steady state) for small regions (defined by degree of latitude and productivity class for dominant species) as attempts for high precision of the estimates applied for the period of the last few thousands of years are uncertain due to high variation of data and factors affecting plot history.*

*For an improved understanding of the biomass models we reformulated Section 2.1.1 and Appendix C (see the marked up version of our BGD paper in the supplement of this comment). We also replaced Fig. B1 and S9 by Fig. 2 and 3 of this comment and added the component biomass and litter contribution Fig. 1 into the supplement as Fig. S10.*

*We noticed the erroneous unit in the original caption of Fig. B1 where the units "$\text{tons ha}^{-1}$" in scatterplots of the non-linear models were instead described as "$\text{kg ha}^{-1}$". We have redrawn Fig. B1 and S9 using "$\text{tC ha}^{-1}$" (Fig. 2) and added $R^2$ values.*

*Interestingly your comments on validity of our understory models complemented on previous comments from Prof Göran Ågren who was interested whether our stand biomass models based on $f_{APAR70}$ accurately reflected Swedish regional differences in nutrient status and Nitrogen deposition (as possible reason for biased estimates of SOC stock on fertile sites). Note,that based on Prof Göran Ågren comments we have redrawn Fig. A1 and added new Fig. S11 in the supplement of the BGD paper. You are most welcome to interact with Prof Göran Ågren and us replying to him on the discussion page of our paper.* http://www.biogeosciences-discuss.net/bg-2015-657/

**It is not clear what authors wanted to show by this manuscript. From the presented results it follows that models of some processes do not accurately reflect these real processes. But it is evident and not new! Another conclusion of the article is also obvious: data for model essentially impact the model results.**

*# In the view of the above mentioned general conclusions, we (1) clarified the novelty*

*of our study by highlighting the connection between the soil nutrient status and performance of widely applied soil carbon models (see reformulated Conclusions), and (2) mentioned that the use of the long-term mean litter input, instead of using litter from the actual state forest measurements, has mainly contributed for accurate modelling of SOC stocks (see reformulated Section 2.1.1). The second was obviously necessary for accurate analysis and it is not meant to be a conclusion of our study, therefore it was removed from conclusions (see reformulated Conclusions).*

*What we meant to describe in our Yasso07, Q, and CENTURY model intercomparison with Swedish soil carbon inventory data was that process based soil carbon models with the current formulation lacking nutrient status related controls of decomposition and soil carbon accumulation would underestimate for conditions where the high nutrient status predominate, in our application for medium-highly productive sites of Southern Sweden. Thus, the main message of our study is the modelling SOC stock bias related to the application of the Yasso07, Q, and CENTURY soil carbon models on productive sites in Sweden, which have not been published by other scientists and that is new to a wide community of modelers or other users of these models. As mentioned in our BGD paper and described further in detail in above discussion, our simulation is based on the widely used process based SOC models, accurate driving data including litter inputs, and massive SOC data points (Swedish inventory data, N=3230). Through the intercomparison of three different widely-used SOC models with massive data points, we identified that re-evaluating of the impact of nutrient status would improve the model development towards their accuracy on estimation of SOC stocks. Therefore, our study is very useful for developing accurate soil carbon and Earth system models, needed for accurate estimation of feedback of global warming on SOC stock temperature sensitivity and soil CO2 efflux, for the accurate national reporting of soil carbon stock changes for United Nations Framework Convention on Climate Change (UNFCCC), and implications of decisions mitigating the climate change effects on soil carbon stocks.*

*For an improved clarity of the main message we reformulated Conclusions (see the marked up version of our BGD paper in the supplement of this comment).*

**I think this manuscript can not be published**

*# We are aware of your concerns about the low R2 values of our understory biomass models presented in Table C1, and about the clarity of the main message. However, as we thoroughly clarified above, the use of these models in our application is reasonably accurate and does not introduce bias on the estimated SOC stocks of soil carbon models and onto their relations to site nutrient status. In sections describing biomass models, we improved the description of the influence of litter input components onto total litter input and SOC stock results. In above response and in improved conclusions we also highlighted the main message of our study.*

*We hope that you could reconsider this statement after improvements made into the paper, and that if needed you would give us further comments suggesting necessary improvements.*

**FIGURE CAPTIONS:**

**Fig. 1. (Fig. S10.)** The tree stand and understory forest (a) biomass, (b) litterfall, and (c) understory litterfall (all in $tC\,ha^{-1}$) for Swedish Forest Inventory plots with available understory coverage observations and in their actual state close to the estimated long-term mean conditions "steady state".

**Fig. 2. (Fig. B1.)** Scatter plots for the dry weight tree biomass components ($tC\,ha^{-1}$) between "modelled" (estimated based on fraction of absorbed radiation, $f_{APAR}$, and our $f_{APAR}$ models) and "measured" (estimated based on basic tree stand dimensions and allometric biomass models). The $r^2$ values represent the coefficient of determination indicating how close the modeled values fit the measured values.

[Figure]

**Fig. 3. (Fig. S9.)** Scatter plots for the dry weight biomass ($\mathrm{tC\,ha}^{-1}$) of the functional types of understory vegetation for Swedish Forest Inventory plots in actual state being close to the estimated long-term mean conditions "steady state". On the x-axis is the biomass modelled by the understory vegetation dry weight biomass ($\mathrm{tC\,ha}^{-1}$) models and on the y-axes is the observed coverage multiplied by the coverage/biomass conversion functions. The abbreviations "abv", "belw", and "tot" mean aboveground, belowground and total. The last panel for "understory total" shows high agreement between the sums of each modeled functional types and the sums of all functional types. The $r^2$ values represent the coefficient of determination indicating how close the modeled values fit the observed values.

Please also note the supplement to this comment:
http://www.biogeosciences-discuss.net/bg-2015-657/bg-2015-657-AC2-supplement.pdf

―――――――――――――――――

[Figure]

[Figure]

[Figure]

**Fig. 1.**

[Figure]

Fig. 2.

[Figure]

**Fig. 3.**

**Supplement:**

[revised manuscript text omitted]
{tC\,ha^{-1}})$ of the functional types of understory vegetation for Swedish Forest Inventory plots in actual state being close to the estimated long-term mean conditions "steady state". On the x-axis is the biomass modelled by the understory vegetation dry weight biomass $(\mathrm{tC\,ha^{-1}})$ models and on the y-axes is the observed coverage multiplied by the coverage/biomass conversion functions. The abbreviations "abv", "belw", and "tot" mean aboveground, belowground and total. The last panel for "understory total" shows high agreement between the sums of each modeled functional types and the sums of all functional types. The $r^2$ values represent the coefficient of determination indicating how close the modeled values fit the observed values.

[Figure]

**Figure S10.** The tree stand and understory forest (a) biomass, (b) litterfall, and (c) understory litterfall (all in $\text{tC ha}^{-1}$) for Swedish Forest Inventory plots with available understory coverage observations and in their actual state close to the estimated long-term mean conditions "steady state".

[Figure]

**Figure S11.** Scatterplots between Latitude (°) and the actual state forest litterfall ($\mathrm{tC\,ha^{-1}\,y^{-1}}$) a), b), c) or long-term mean "steady state" forest litterfall ($\mathrm{tC\,ha^{-1}\,y^{-1}}$), g) h), i); and scatterplots between Nitrogen deposition ($\mathrm{kgN\,ha^{-1}\,y^{-1}}$) and the actual state forest litterfall ($\mathrm{tC\,ha^{-1}\,y^{-1}}$) d), e), f) or long-term mean "steady state" forest litterfall ($\mathrm{tC\,ha^{-1}\,y^{-1}}$) j), k), l) for deciduous species, Scots pine, and Norway spruce dominated stands.

---

## Referee Comment (RC2) · Anonymous Referee #2 · 7 Mar 2016

Summary:

Three soil models (Q, Yasso07 and CENTURY) are ran against Swedish forest soil inventory data to gauge how well they can estimate soil C stocks. The soils were additionally broken down into 10 distinct groupings based on soil characteristics or 5 on site characteristics. Generally the models perform well enough but have problems with certain sites characterized by high fertility and are generally well-sorted for parent material.

I have some troubles with understanding the point of the paper. The authors took three

separate soil C models and ran them then compared them. That is fine but why not have examined how the special characteristics of CENTURY could have helped its performance? The authors noted that CENTURY simulates its soil C to only 20 cm and they noted that it should likely be increased by 40-50% like Yasso07, but then why not show on plots how that would look? Similarly, CENTURY is capable of N dynamics and the authors explicitly note that N deposition at some sites seems to be important, so why not do a run with the N-cycle turned on? Then at least we could see how well the model does when its full capabilities are used. This strikes me as taking a Ferrari, deflating all of its tires, filling it with poor petrol and then racing against a Honda. Sure it's performance can be evaluated but it is hardly ideal conditions to see how fast it can really go.

I also worry about the litter inputs. I would have liked to see some way of independently evaluating the litter input contributions.

I recommend the authors do some further simulations to make this paper more interesting and to offer up a better analysis of how the model processes can contribute to estimated SOC stocks (thinking here the N cycle in CENTURY). I usually don't like to ask for more simulations but in this case I think it is necessary to make the paper have wider appeal. If not a more specialized journal could be appropriate.

Specific comments:

1. The paper is generally not well written and would greatly benefit from English copy-editing. I mention this as I often had to re-read sections to understand what was written. There are a few areas where I still don't understand what was being communicated.

2. The section on fAPAR was hard to follow ('actual state'? I don't understand if this was an English problem or if this term was meant. It is a strange term to be used). In the end I was not sure how good this fAPAR method worked out. I can't see anywhere that this was explicitly tested against some sort of observations. Since the litter inputs are pretty important to drive the models with, shouldn't this be very well evaluated?

[Figure]

3. How was the stump defined for the biomass? Usually I think of stem, coarse roots, and fine roots with the stump being what is left after a site is logged. How was it meant here?

4. Line 264 - But the CENTURY simulation was run to equilibrium, right? Also how was equilibrium defined for the models?

5. Table 2, how is the productivity class derived?

6. Table 2 - The depth of soil is assumedly cut off at 1 meter?

7.Table 3 - Parameters (leftmost column)? What is meant here? How the model was parameterized? I found this confusing.

8. Table 3 - CENTURY, is the soil depth adjustable from 0.2? Could it be increased to 1.0 to more simply make it comparable to the other models?

9. Figure 2 and text in main - Soil group 8 has only 8 samples within it. Is this reasonable to keep as a group? Given how many uncertainties develop as this regression tree is created (calculation of SOC, assignment to weather stations, measurement uncertainty, etc.) is it reasonable to let a group be only 0.24% of the total?

---

## Author Comment (AC3) · 9 Apr 2016

**Referee's comments are highlighted by the bold font.**
*Author's replies are indicated by the italic font.* The normal font indicates text of the manuscript.

**Review of Tupek et al.**
**Summary:**
**Three soil models (Q, Yasso07 and CENTURY) are ran against Swedish forest soil inventory data to gauge how well they can estimate soil C stocks. The**

[Figure]

**soils were additionally broken down into 10 distinct groupings based on soil characteristics or 5 on site characteristics. Generally the models perform well enough but have problems with certain sites characterized by high fertility and are generally well-sorted for parent material.**

*Thank you for your comments! We appreciate and considered them all, and below we reply to each in detail. Based on your comments we have resimulated CENTURY SOC stocks with tuned parameters accounting for the variation of topsoil mineral N, C/N ratio of litterfall in relation to site N deposition and productivity class. We redrawn Fig. 3 and 4, supplement Fig. S5, S7, and S8, and added Fig. S10, S11, and S12. We reformulated text by following your remarks and according to improved performance of CENTURY model (as in the marked up version of the manuscript attached into the supplement of this comment).*

**I have some troubles with understanding the point of the paper.**

*The point of the paper was evaluating Yasso07, Q, and CENTURY model estimates of SOC stocks wheather they can follow the variation of measured SOC stocks when those were grouped according to site nutrient status (Fig. 3 of the BGD manuscript), and helping to understand why models performed well for 2/3 of sites and failed for more fertile sites. We reformulated Conclusions.*

**The authors took three separate soil C models and ran them then compared them. That is fine but why not have examined how the special characteristics of CENTURY could have helped its performance?**

*We presented our model intercomparison keeping some special CENTURY character-isitcs constant, because we included the main driver of these models, litter input, and did not acount for all drivers in CENTURY as we expected them to have small effect on estimated SOC stocks. We have now confirmed by CENTURY sensitivity simulations that in comparison to litterfall including parameters of topsoil mineral N, and C/N ratio of the litterfall had small effect on SOC stocks (Fig.1).*

**The authors noted that CENTURY simulates its soil C to only 20 cm and they**

**noted that it should likely be increased by 40-50 % like Yasso07, but then why not show on plots how that would look?**

*We did not scale CENTURY estimates because we were interested more in reproducing the pattern of the grouped measurements. Scaling the topsoil horizon SOC stock by adding 40% of estimated site specific SOC stock to account for the deep carbon in the current version of the manuscripts (described in section 2.2) helped CENTURY estimates to agree with measurements, thus in the current version of the manuscript we presented scaled CENTURY SOC stocks.*

**Similarly, CENTURY is capable of N dynamics and the authors explicitly note that N deposition at some sites seems to be important, so why not do a run with the N-cycle turned on? Then at least we could see how well the model does when its full capabilities are used. This strikes me as taking a Ferrari, deflating all of its tires, filling it with poor petrol and then racing against a Honda. Sure it's performance can be evaluated but it is hardly ideal conditions to see how fast it can really go.**

*We noticed that part of our BGD manuscripts discussion on line 464, in particular that* "...the feedback of nitrogen input to plant productivity was not included in this study" *was misleading and has to be reformulated into* "...the feedback of nitrogen input to plant productivity was primarily included in this study indirectly, through estimated steady state litter input based on site productivity class which strongly correlated with Nitrogen deposition (Fig. A1 and S11)."

*As litter input indirectly reflected N deposition, we focused on C part of CENTURY (that is common by modelling studies) by accounting for the main drivers of SOC stocks sequestration site specific litter input, climate, and soil texture and structure. Although in our BGD manuscript we did not presented the results of CENTURY soil sub-model in its full capabilities, in the current version of the manuscipt (Section 2.2) we further accounted for N part through the contribution of site specific parameters of topsoil mineral N (relative to N deposition, Throop et al. 2004), C/N ratio of the litterfall (relative to production, Merilä et al. 2015), and we also included effect of drainage*

*(relative to long-term soil moisture, Raich et al. 2000) (Fig. 2). We also found and corrected a mismatch between the site specific soil silt-, clay-, sand- contents and other input data (correctly used litterfall and climate) that caused under-performance of CENTURY in our BGD manuscript. After soil input data were matched correctly with the other input data, then the CENTURY SOC stock estimates improved into more pronounced spatial (between group) differences. The CENTURY estimates were markedly larger for the groups with higher clay contents and generally lower for the other groups (Fig. 2).*

Throop, H. L., Holland, E. A., Parton, W. J., Ojima, D. S. and Keough, C. A.: Effects of nitrogen deposition and insect herbivory on patterns of ecosystem level carbon and nitrogen dynamics: results from the CENTURY model, Global Change Biol., 10, 1092-1105, 2004.

Merilä, P., Mustajärvi, K., Helmisaari, H., Hilli, S., Lindroos, A., Nieminen, T. M., Nöjd, P., Rautio, P., Salemaa, M. and Ukonmaanaho, L.: Above-and below-ground N stocks in coniferous boreal forests in Finland: Implications for sustainability of more intensive biomass utilization, For. Ecol. Manage., 311, 17-28, 2014.

**I also worry about the litter inputs. I would have liked to see some way of independently evaluating the litter input contributions.**

*The main driver of the SOC stock accumulation, the forest plant's litterfall, was precisely estimated based on the ground measurements of Swedish forest inventory data and Scandinvian biomass and litterfall functions, and for the main Swedish regions agreed with Ortiz et al. (2013). The developed functions based on $f_{APAR}$ were through removing the effect of the management (the present stand development) the main contributors for accurate estimation of the long-term mean litter input (newly added Fig. S11 in the supplement of the edited manuscript). The allometric biomass models used to derive our $f_{APAR}$ biomass models were based on studies using extensive data from boreal forest of Scandinavia (lines 133-134). The biomass estimates of the published allometric functions and our $f_{APAR}$ functions strongly correlated ($R^2$*

*values close to 0.9, Table B1 and Fig. B1). Litterfall estimation as a proportion of forest biomass was also based on studies from Scandinavia (lines 153-165) and our estimates of litterfall components of steady state forests (newly added Fig. S10 in the supplement of the edited manuscript) were within the range of reported values (Ågren et al. 2007, Mukkonen and Lehtonen 2004, Lehtonen et al. 2004, Viro 1955, Mälkönen 1974, 1977, Kleja et al. 2008, Leppälampi-Kujansuu et al. 2014, Liski et al. 2006, Ortiz et al. 2013). For an improved understanding of the $f_{APAR}$ biomass models we reformulated Section 2.1.1, Appendices A and B, redrawn Fig. A1 and B1, and added supplement Fig. S10 and S11. The appendix Fig. A1 was redrawn in order to increase clarity of biomass/litterfall modelling based on the productivity class, and supplement Fig. S10 shows the range of litter input, Fig. S11 increases clarity of biomass/litterfall modelling on the Nitrogen deposition.*

**I recommend the authors do some further simulations to make this paper more interesting and to offer up a better analysis of how the model processes can contribute to estimated SOC stocks (thinking here the N cycle in CENTURY). I usually don't like to ask for more simulations but in this case I think it is necessary to make the paper have wider appeal. If not a more specialized journal could be appropriate.**

*In the current version of the manuscipt (in supplement of this comment) we present results from the tuned CENTURY model that includes site specific parameters of topsoil mineral N, C/N ratio of the litterfall, and drainage. However, tuning of CENTURY parameters to site specific topsoil mineral Nitrogen, C/N ratio of the litterfall, and drainage (Fig. 1 and Fig. 2) showed that this impact on SOC stocks estimates was small in comparison to sensitivity of SOC stock estimates to litterfall. The Fig. 1a showed that 30% increase in litterfall increased SOC stocks by 15 $\mathrm{tC\,ha^{-1}}$, whereas tuning the parameters of C/N ratio of litterfall by 30% resulted only in SOC stock change up to 1 $\mathrm{tC\,ha^{-1}}$ (Fig. 1b) and increasing mineral N by 30% increased estimates up to 2 $\mathrm{tC\,ha^{-1}}$ (Fig. 1c). Further increase of topsoil mineral N resulted to maximum*

*SOC stock increase around 5 tC ha$^{-1}$ compared to setting used in our BGD manuscript (Fig. 1c and 1d). The Fig. 1 and Fig. 2 showed that litterfall was the main driver of the estimated SOC stocks and therefore accurate SOC stocks depended on accurate biomass and litterfall estimation.*

*We added description of the CENTURY simulation with N, C/N and drainage parameters into the manuscript (section 2.2.), added Fig.1 to supplement as Fig. S12, and redrawn the figures containing CENTURY estimates. Although the main message of the edited manuscript remained similar to the previous version, we reformulated our findings regarding the improved performance of the CENTURY model and conclusions.*

**Specific comments:**

**1. The paper is generally not well written and would greatly benefit from English copyediting. I mention this as I often had to re-read sections to understand what was written. There are a few areas where I still don't understand what was being communicated.**

*English language of our BGD manuscript was revised by a native speaker. For the additional clarity we reformulated mainly sections 2.1.1 and conclusions. Manuscript in final form would undergo English copyediting services.*

**2. The section on fAPAR was hard to follow ('actual state'? I don't understand if this was an English problem or if this term was meant. It is a strange term to be used). In the end I was not sure how good this fAPAR method worked out. I can't see anywhere that this was explicitly tested against some sort of observations. Since the litter inputs are pretty important to drive the models with, shouldn't this be very well evaluated?**

*We reformulated section 2.1.1 for increased clarity between the actual and steady state forest, and the use of $f_{APAR}$ models. We meant to use the term 'actual state'*

*referring to current state, existing at the present time, as used to describe phenomena in physics. However, our focus was not on the actual state, but on the long-term mean conditions what we referred as 'steady state'.*

*Our use of $f_{APAR}$ models for steady state was motivated by the need to remove the effect of management from the Swedish Forest Inventory measurements and to produce biomass/litterfall estimates accurately representing the mean long-term conditions (defined by estimated steady state) for small regions (defined by degree of latitude and productivity class for dominant species) (see redrawn Fig. B1). The higher precision of the estimates applied for the period of the last few thousands of years would be uncertain due to high variation of factors affecting plot history. As shown by Fig. S11 the litterfall based on $f_{APAR}$ models of steady state forests were sensitive to regional differences in N deposition that correlated to site productivity, and estimated litterfall components (Fig. S10) were in agreement of studies from Scandinavia.*

**3. How was the stump defined for the biomass? Usually I think of stem, coarse roots, and fine roots with the stump being what is left after a site is logged. How was it meant here?**

*#The stump was defined and calculated as a difference between the felled part of the tree and roots that were attached to it (Pettersson and Ståhl 2006, lines 131-134).*

Petersson, H. and Ståhl, G.: Functions for below-ground biomass of Pinus sylvestris, Picea abies, Betula pendula and Betula pubescens in Sweden, Scand. J. For. Res., 21, 84-93, 2006.

**4. Line 264 - But the CENTURY simulation was run to equilibrium, right? Also how was equilibrium defined for the models?**

*The equilibrium state of a model was a state where the litter input equals decomposition and it is referred as the steady state soil carbon stock (described on lines 224-225 for Q, 235 for Yasso07 , and 262-264 for CENTURY models).*

**5. Table 2, how is the productivity class derived?**
*We added following sentences into section 2.1.3 of edited manuscript:*
The productivity class (H100, m) in our manuscript refers to a site index which can be converted to site productivity. Soil site index is based on dominant height at a certain age (100 years) and is determined according to a dominant height curve (Swedish Statistical Yearbook of Forestry 2014).
Swedish Statistical Yearbook of Forestry. Official Statistics of Sweden, 370 p., Skogsstyrelsen. 2014.

**6. Table 2 - The depth of soil is assumedly cut off at 1 meter?**
*Yes, the SOC stock represented the soil assumedly cut off at 100 cm (Stendahl et al. 2010). We added this information into the header of Table 2.:*
The soil was cut off at 1 meter.

**7.Table 3 - Parameters (leftmost column)? What is meant here? How the model was parameterized? I found this confusing.**
*#Parameters (leftmost column of Table 3) used in models represented different scales. Yasso07 parameters were global, Q parameters were regional (Scandinavian), and CENTURY parameters were combination of global and site specific for soil and C/N ratio of litterfall. We reformulated this line of Table 3 as:*
Parametrization: Global, Scandinavian, Global and site specific.

**8. Table 3 - CENTURY, is the soil depth adjustable from 0.2?Could it be increased to 1.0 to more simply make it comparable to the other models?**
*We added following sentence into the section 2.2. of edited manuscript:*
In order to account for the deep soil carbon (Jobbágy and Jackson 2000), we scaled CENTURY estimates representing the topsoil horizon by adding 40% of estimated site specific SOC stock.
Jobbágy, E. G. and Jackson, R. B.: The vertical distribution of soil organic carbon and its
relation to climate and vegetation, Ecol. Appl., 10, 423-436, 2000.

**9. Figure 2 and text in main - Soil group 8 has only 8 samples within it. Is this reasonable to keep as a group?Given how many uncertainties develop as this regression tree is created (calculation of SOC, assignment to weather stations, measurement uncertainty, etc.) is it reasonable to let a group be only 0.24% of the total?**

*The soil group 8 that has only 8 samples was in our opinion distinct from the others as found by the rpart (Fig. 3). We added following sentences into the section 2.1.3:*
We acknowledge the fact that this is a small distinct group based only on 8 observation. However, we don't have any reasons to exclude these datapoints as outliers.

**FIGURE CAPTIONS:**
**Fig. 1. (Fig. S12)** Sensitivity of simulated SOC stocks ($tC\,ha^{-1}$) of CENTURY model to variation in litterfall (a), C/N ratio of litterfall (b), topsoil mineral N ($gN\,m^{-2}$) (c), and to variation of factors together (d). SOC stocks of CENTURY are output of spin up simulation up to 1000 years.

**Fig. 2.** Bean plot of density functions for 10 physicochemical groups of the soil carbon ($tC\,ha^{-1}$) measurements (grey fill) and estimates simulated by the soil carbon models Yasso07, CENTURY, and CENTURY tuned (including site specific mineral N in topsoil, C/N ratio of litterfall, and drainage), Q with the litter input derived from the steady state forest. The thin lines are the density distributions. The thick lines are the group means and dashed lines are their confidence intervals. The n is number of samples. For description of group levels of SOC stocks, moisture, and fertility see Fig.2 and Table S1. *Note that in the edited manuscript (Fig. 3) we show CENTURY estimates including all used parameters (tuned), in order to keep balance with the results of Yasso07 and Q models.*

Please also note the supplement to this comment:
http://www.biogeosciences-discuss.net/bg-2015-657/bg-2015-657-AC3-
supplement.pdf

―――――――――――――――――

[Figure]

**Fig. 1.**

[Figure]

**Fig. 2.**

**Supplement:**

[revised manuscript text omitted]
 $(\text{tC ha}^{-1})$ of the functional types of understory vegetation for Swedish Forest Inventory plots in actual state being close to the estimated long-term mean conditions "steady state". On the x-axis is the biomass modelled by the understory vegetation dry weight biomass $(\text{tC ha}^{-1})$ models and on the y-axes is the observed coverage multiplied by the coverage/biomass conversion functions. The abbreviations "abv", "belw", and "tot" mean aboveground, belowground and total. The last panel for "understory total" shows high agreement between the sums of each modeled functional types and the sums of all functional types. The $r^2$ values represent the coefficient of determination indicating how close the modeled values fit the observed values.

[Figure]

**Figure S10.** The tree stand and understory forest (a) biomass, (b) litterfall, (c) stand litterfall and (d) understory litterfall (all in $\mathrm{tC\,ha^{-1}}$) for Swedish Forest Inventory plots with available understory coverage observations and in their actual state close to the estimated long-term mean conditions "steady state".

[Figure]

**Figure S11.** Scatterplots between Latitude (°) and the actual state forest litterfall ($tC\,ha^{-1}\,y^{-1}$) a), b), c) or long-term mean "steady state" forest litterfall ($tC\,ha^{-1}\,y^{-1}$), g) h), i); and scatterplots between Nitrogen deposition ($kgN\,ha^{-1}\,y^{-1}$) and the actual state forest litterfall ($tC\,ha^{-1}\,y^{-1}$) d), e), f) or long-term mean "steady state" forest litterfall ($tC\,ha^{-1}\,y^{-1}$) j), k), l) for deciduous species, Scots pine, and Norway spruce dominated stands.

[Figure]

**Figure S12.** Sensitivity of simulated SOC stocks $(\mathrm{tC\,ha^{-1}})$ of CENTURY model to variation in litterfall (a), C/N ratio of litterfall (b), topsoil mineral N $(\mathrm{gN\,m^{-2}})$ (c), and to variation of factors together (d). SOC stocks of CENTURY are output of spin up simulation up to 1000 years.

---

## Author Response (AR3)

**Author's reply on "Underestimation of boreal soil carbon stocks by mathematical soil carbon models linked to soil nutrient status" by B. Ťupek et al. 6[th] July 2016**

boris.tupek@luke.fi

Referee's comments are highlighted by bold font.
**Symbol and font used to indicate Author's reply.**

**Anonymous Referee #3:**
**General comments**
**The paper Ťupek et al. is aimed to test the ability of some biogeochemical models (Yasso07, Q, and CENTURY) to predict the soil organic carbon stocks in Swedish forests. For model validations a large set of forest and soil inventory data collected in different forest regions in Sweden are used. As one of the key results of the study it was shown that the models are not able to predict adequately the soil carbon accumulation in forest sites with very high nutrient status. The question about model ability to predict the soil carbon stock under over-moist soil conditions (high ground water level) was also discussed.**
**The main goal and objectives of the study is corresponded to the main scope of Biogeoscience journal and the paper can be published in BGS after some revision.**
*#Thank you! We appreciate your comments and below we provide a point by point response to each of them.*

**Before publication of the manuscript the several points have to be additionally clarified and discussed.**
**It is not completely clear why the authors use in the study "an actual fraction of photosynthetically active absorbed radiation (fAPAR) as a relative indicator of a site's capacity to produce biomass"?**
*#We clarified the reason why to use fAPAR in 2.1.1 Biomass and litterfall estimates.*
We adopted the actual fAPAR as a relative indicator of a site's capacity to produce biomass (minimum = 0, maximum = 1) by accounting for the forest stand structure, ranging from the absent stand fAPAR = 0 to the closed canopy stand fAPAR = 1, through its major role on limiting of the potential gross primary production (e.g. Peltoniemi et al. 2015). Peltoniemi M., Pulkkinen M., Aurela M., Pumpanen J., Kolari P. & Mäkelä A. 2015: A semi-empirical model of boreal-forest gross primary production, evapotranspiration, and soil water — calibration and sensitivity analysis. Boreal Env. Res. 20: 151–171.

**Authors write that "the fAPAR was calculated using basic tree measurements".**
*#We reformulated confusing expression of "basic tree measurements" to "SFI measurements of basic tree dimensions" that were explained in first sentence of the section 2.1.1 Biomass and litterfall estimates.*

**And from fAPAR values the forest productivity is estimated. Application of such indirect approach to estimate the forest productivity within the frameworks of the study is not clear. Actually the amount of above ground photosynthesizing biomass that characterizes the forest productivity can be either estimated directly from the forest inventory or derived from remote sensing data (e.g. from NDVI).**
*#Yes, we initially estimated the actual forest photosynthesizing biomass and litter from the forest inventory. However, these "observed - estimated directly from SFI" actual biomasses and litter values include information of forest management that masks the effect of site productivity and nitrogen deposition (see Fig. S11). Rather than using the observed snapshot of the history, more realistic litter input to the models for thousands of years has to be based on an average long term biomass/litter estimates.*

*The average long term forest biomass corresponds to the proportion of the maximum "observed" biomass for the small regions outlined by latitude, productivity class and dominant tree species distribution. Instead of modelling the forest biomass components maximum for a latitudinal degree, site productivity class, and tree species, and finding its proportion, it was simpler to estimate maximum fAPAR and sought its fraction iteratively based on the distributions of measured and modeled soil organic carbon stocks (Fig. S2). (Appendix A).*

*We added following section into the manuscript in 2.1.1 Biomass and litterfall estimates:*
Instead of modelling of average long term biomasses for every tree stand component separately for the species, latitude, and productivity index, we simplified the biomass modelling firstly by estimating only a long term forest stand structure for the species, latitude, and productivity (fAPAR70, Table A1) and secondly by using fAPAR70 with fAPAR biomass models (Table B1) to estimate the biomass components.

**It is clear that the fAPAR approach is broadly used in remote sensing to quantify the GPP and NPP of land surface in regional and global scales. However GPP and NPP estimations on local scale using this approach are usually very uncertain. Dependences of NPP and GPP on fAPAR even for growing non-damaged forest stands are often non-linear. The fAPAR is actually depended on many different parameters including amount of above-ground green biomass, amount of non- photosynthesizing biomass, forest understorey, soil properties including composition, texture and moisture, etc., etc**

*# As above mentioned, in our study the site specific fAPAR was directly calculated based on Swedish forest inventory data and a method described in Härkönen et al. (2010).* Härkönen, S., Pulkkinen, M., Duursma, R. and Mäkelä, A.: Estimating annual GPP, NPP and stem growth in Finland using summary models, For. Ecol. Manage., 259, 524-533, 2010.

*The fAPAR values in our study represented the observed state of forest stand biomass.*

**Other point. Authors use the tree height as an indicator to quantity the forest "site" productivity. The forest productivity is actually depended on forest species composition, influenced by the air temperature, precipitation, soil properties, water and nutrient supply during the previous years, etc. Nowadays it is very popular to use the tree height as some relative indicator of forest productivity. However it is relatively rough indicator. In many cases the tree of different species reaches the maximal heights some time before the reference age used by authors - 100 years. In this case the tree height is not the best characteristics for forest productivity. The productivity (in particular aboveground production) of the trees for the time from the stage of seedlings to stages of mature or old trees can be actually characterized by two main characteristics - tree height and tree steam diameter (BHD). So it is better actually to use combined indicator for forest productivity including the tree height and BHD.**

*# We agree that there are many kinds of indicators of forest productivity and tree height is one of the most widely used. However, site index H100 used in our study was comprised in Swedish forest inventory data and it was estimated not only based on tree height, but also based on specific site properties (Hägglund and Lundmark, 1977).*

*We added following section into the manuscript:*

The site index (H100, dominant height at a total age of 100 years), that can be translated to a specific productivity ($m^3 ha^{-1} yr^{-1}$), was in our study calculated for sites based on observed site properties from Swedish forest inventory by using the methodology of Hägglund and Lundmark (1977) (Swedish Statistical Yearbook of Forestry 2014). Hägglund, B., Lundmark, J.-E., 1977. Site index estimation by means of site properties. Scots pine and Norway spruce in Sweden. Studia Forestale Suecica 138, 1–38.

**In figure A1 I suggest to replace the productivity (title of X axe) by tree height. To avoid any misunderstanding.**

*#we replaced "productivity" in Fig. A1 by "site index H100"*

**Specific comments**
**Line 3. "... carbon exchange ...". Do you mean the total carbon or CO2 only?**
*# we mean CO2 and changed "carbon" in the text to* "carbon dioxide"
**Line 26. Do you mean the carbon dioxide?**
*#we mean CO2 and changed "carbon" in the text to* "carbon dioxide"
**Line 331 Annual temperature cannot be never "cold". Cold winter, but the temperature is low or high.**
*# the word "cold" was on line 331 replaced by "low"*
**Significance level (p=value) in Fig.4, Tab.A1, B1, C1 has to be indicated.**
*#we added p values to Fig. 4, in Tab. A1, B1, C1 p values are indicated in the footnotes.*
**Table A1, B1. Please to round the numbers to 1 or 2 digits after comma.**
*#we rounded the numbers to 2 digits in Table A1 and in Table B1*

**Anonymous Referee #2:**
**Thank you for your careful considerations of my comments from the first round however I have some of the same concerns from the first round and a few further points.**
*#Thank you!*

**1. When I expressed confusion about the use of the term 'actual state' that should be a hint that maybe it is not a good term. It may be very common in physics, but this is not a physics journal, nor is the audience going to be primarily physicists. Use the language of your audience. Consider changing actual state to be 'observed' if the quantity is indeed observed. It is more clear then the derivation of the information. Just because it is the state at the present time doesn't tell me if it is an observed or modelled quantity. Similarly, I would use equilibrium value rather than steady-state. Again the language of your readers...**
*#we changed the terms as required "actual state" to "observed" and "steady state or long-term mean" to*

*"equilibrium" in all instances*

**2. Why did none of the work with 'tuned' CENTURY not make it into the revised MS and only in the response to reviewers? That should go in the MS.**
*#We are sorry for the misunderstanding on the content of revised MS, please note that we already described the "tuned" CENTURY in the revised manuscript on lines 278-288 and also included these results into our model comparison (Fig. 3, Fig. 4, and Fig. S12, Section 3 and 4).*

**3. I don't agree that keeping soil group 8 with only 0.24% of points is valid. Either show that it is not just an artifact or error from your regression tree or remove it. It is obviously very strange given how much it differs from the other groups and how far away the models are. If it is a real group and not just sampling error or assignment error it should be better justified. There are 5 decisions steps to get to that group, prove that you didn't make a mistake on one of those five. Conversely really try to answer why that group is so special and look into the measurements to understand the conditions of those few sites better.**
*#We still think that including data of the smallest regression tree group 8 is valid and well justified in the manuscript (Fig. 2, Fig. S2, Fig. S4, Table S1, and their descriptions in Section 3 and 4), and that the relatively small number of points and a large difference from the other groups is no reason for excluding these data. As requested, we show in the table below that data of the smallest regression tree group 8 correctly represent the unique conditions as shown in Fig. 2a. Furthermore, comparing table below and the description statistics of this group in Table S1 indicated highly fertile conditions (large N deposition, H100 is largest among groups (31 m), second largest litter input, highest temperature, and highest precipitation, on well-drained soil). Please note, that in the table below and in Fig. S4, that SOC stocks of mineral soil are extremely large indicating fast soil organic matter dynamics and microbial transformation, SOC transport downwards and its stabilization, and confirming MEMS hypothesis of Cotrufo et al. (2013) and one of our conclusions that mechanisms of mineral soil associated SOC stabilization of process based soil carbon models has to be re-evaluated.*

| North (°) | East (°) | SOC soil (t/ha) | SOC humus (t/ha) | SOC total (t/ha) | CEC.BC (mmolc/kg) | CN.BC | N deposition (kgN/ha/year). | bound.H2O.C ( %) |
|---|---|---|---|---|---|---|---|---|
| 56.26 | 15.27 | 204.28 | 19.60 | 223.88 | 28.92 | 21.36 | 10.35 | 3.05 |
| 56.96 | 13.09 | 211.71 | 24.65 | 236.35 | 29.29 | 21.50 | 16.15 | 2.91 |
| 57.11 | 12.27 | 202.23 | 67.87 | 270.11 | 17.90 | 25.53 | 16.56 | 11.42 |
| 56.96 | 13.09 | 285.92 | 120.32 | 406.24 | 29.76 | 33.62 | 17.82 | 5.43 |
| 56.95 | 13.06 | 310.38 | 16.70 | 327.08 | 22.71 | 19.17 | 17.63 | 2.63 |
| 56.03 | 12.77 | 255.48 | | 255.48 | 24.71 | 25.27 | 17.19 | 2.69 |
| 56.95 | 13.06 | 177.35 | 36.57 | 213.92 | 17.54 | 21.71 | 18.52 | 2.53 |
| 56.96 | 13.09 | 197.47 | 18.36 | 215.83 | 25.43 | 16.22 | 14.95 | 3.19 |

*coordinates in the table correspond to the closest weather station and not to an exact location of soil inventory plots

**4. Fig B1, add the number of points on each (n=...) so us readers have that info.**
*#We added n (number of points) into the Fig. B1.*

**5. I see now that Fig A1 caption no longer calls it 'measured fAPAR' but 'actual fAPAR'. So that seems to mean that we have 'actual fAPAR' which is a calculated quantity based on Harkonen et al. and your modelled fAPAR (?). So really we are comparing model vs. model? So where do things like observations fit in?**
*# In Fig. A1 caption and through the manuscript the site specific "actual = observed" fAPAR was directly calculated based on observations from Swedish forest inventory data and a method described in Härkönen et al. (2010). Thus fAPAR values in our study represented the observed state of forest stand biomass. The long term average fAPAR (fAPAR70) was found iteratively from the "observed" fAPAR maxima for the sites across Sweden by comparison of SOC stocks estimated with litterfall derived by fAPAR models for fAPAR70 and similarity of distributions between measured and modeled SOC stocks (Fig. S2).*

**This brings me to a more major point. There is a heavy reliance on models to check models within this paper. For eg."We modelled the steady state biomass by applying the fitted exponential functions between the actual state forest biomass components (stem, branch, foliage, stump, coarse-roots, fine-roots, estimated by tree stand measurements and the allometric biomass functions) and the actual fraction of absorbed radiation (fAP AR ) (Appendix B) to the estimated fAP AR70 of the steady state forest" (line 146 in revised MS)**
**and then:**
**"Forest stand biomass was estimated by allometric biomass functions for stem with bark, branch,foliage,**

**stump, coarse-roots and fine-roots applied to basic tree dimensions (breast height diameter,total height of tree, number of trees) of SFI stands " (line 130)**
**Then in appendix B:**
**"The biomass components derived with allometric models (measured) and those derived with fAP AR models (modeled) showed strong correlations (Fig. B1). "**

**But really they aren't measured! They are both modelled quantities; allometric models are not 'real'.This seems like a rabbit hole where modelled quantities are used to check other modelled quantities which then generate further modelled quantities. How do you propagate error from the allometric relations? How do you know that using one modelled quantity to check another results in a correct answer? Since those are very hard questions to answer, it needs to be made completely transparent what is observed and what is modelled. So if you are comparing a modelled value to a modelled value there can be no mistaking it for measured quantities. What you have done could be valid but it is still very opaque on what is going on.**
*# We acknowledge the point needed to clarify our method and test the model outputs with litterfall data. Although data on required litterfall components was not available for Sweden and for soil carbon modelling litterfall needs to be modeled (Ortiz et al. 2011, and 2013), the use of allometric models with data from forest inventory and applying litterfall production rates in order to estimate litter input for soil carbon models is a standard method of national greenhouse gas inventories estimating soil carbon stock changes (Statistics Finland 2013). Therefore we adopted this standard method. In order to model SOC stocks of forest in equilibrium (not SOC stocks changes) we modified it only by estimating the long term litterfall of forest in equilibrium.*

*We added following sentence into the section on 2.1.1 Biomass and litterfall estimates:*
For the biomass and litterfall estimation we adopted standard method of national greenhouse gas inventories for estimating soil carbon stock changes (Statistics Finland 2013). In order to model SOC stocks of forest in equilibrium (not SOC stocks changes) we modified the method by estimating the long term litterfall of forest in equilibrium. Statistics Finland 2013. Greenhouse gas emissions in Finland 1990–2011. In: *National Inventory Report to the UNFCCC Secretariat*, Ministry of the Environment, Helsinki, Finland, pp. 285–286.

**Dr. Leonid L. Golubyatnikov, Referee #1:**
**I do not see in this manuscript new ideas and new approaches. The main results are expected in advance. I think this manuscript does not correspond to the level offered for articles in journal Biogeosciences.**
*# Thank you for your opinion! Although as we thoroughly replied to your previous comments (see* Interactive comment on Biogeosciences Discuss., doi:10.5194/bg-2015-657, 2016), *we do not see any scientifically valid reason for ignoring our results from the SOC stock inter-comparison between the massive Swedish forest soil inventory data and Yasso07, CENTURY and Q process based models.*

*In a nutshell, our study provided unique solution for combining massive ground measurements from Swedish forest and forest soil inventories, estimating litterfall for forest in equilibrium required for modelling equilibrium soil carbon stocks, as well as unique evaluation of physicochemical soil, climatic, and plant factors by using recursive partitioning method for generating regression trees of SOC stocks.*

*We agree that soil nutrient status as a missing driver of soil carbon sequestration of process based models can be hypothesized, as it has been discussed (Schmidt et al., 2011; Averill et al., 2014), and as it's known that soil nutrient status drives the ecosystem carbon use efficiency and soil carbon sequestration (Ågren et al., 2001, Manzoni et al., 2012, Fernández-Martínez et al., 2014), and also that models fail to reconstruct the spatial variation (Todd-Brown et al., 2013; Ortiz et al., 2013; Lehtonen et al., 2015a). Although to our knowledge this has not been tested by other studies, and our findings that Yasso07, CENTURY and Q SOC stock estimates generally agreed well for 2/3 of the Swedish boreal forest sites but underestimated for sites with higher fertility due to models poorly predicting the large carbon stabilization in the mineral soil are new.*

*Biogeosciences (BG) frequently publishing results on interactions between the drivers of soil carbon sequestration provides the best platform for sharing our findings with the international research community. Hopefully by now this has convinced you, that our findings would appeal to a broad audience of scientists studying the interactions between climate, forest stand, soil, and carbon sequestration, such as the readership of BG.*

[revised manuscript text omitted]

---

## Author Response (AR4)

**Author's reply on "Underestimation of boreal soil carbon stocks by mathematical soil carbon models linked to soil nutrient status" by B. Ťupek et al. 12th July 2016**

boris.tupek@luke.fi

Dr. Alexey V. Eliseev, Editor:
Dear authors,
 please make the following technical corrections before final acceptance of the paper:

*#Dear Dr. Alexey V. Eliseev,*
*thank you for the technical corrections which we followed as requested.*

 - l.90: 'into' -> 'until it reaches'
 - l.167: insert the matching bracket after 'TR'
 - l.311: insert 'SOC for the' between 'The' and 'remaining'
 - l.324: change the first letter in 'nitrogen' to lowercase
 - l.577: 'out of' -> 'from the'
*#we changed text as requested*

 - ll.579-583: please reformulate the sentence starting from 'The fAPAR70...' - it is very difficult to read
*#we reformulated text:*
The $f_{APAR70}$ values specific for pine, spruce, and deciduous stands were modeled with latitude and site productivity index (H100) in two steps. Firstly, the $f_{APAR70LAT}$ and the $f_{APAR70H100}$ values were modeled separately by regression models with latitude and with site productivity index (H100) (Table A1). Secondly, the $f_{APAR70LAT}$ was reduced by the difference between the $f_{APAR70H100}$ and the maximum $f_{APAR70H100}$ ($f_{APAR70}$ = $f_{APAR70LAT}$ + $f_{APAR70H100}$ - maximum $f_{APAR70H100}$). The $f_{APAR70}$ equaled the $f_{APAR70LAT}$ only for the maximum site productivity index, otherwise it was reduced.

 - l.609: please explain 'AIC'
*#we added:*
Akaike's information criterion (AIC)

 - Table A1, columns 'a+-SE' and 'b+-SE', sect. 'fAPAR70LAT', lines 'pine' and 'spruce': The number of digits is excessive. Please leave only the digits before the decimal points;
 - Table B1, columns 'a+-SE' and 'b+-SE', all lines: the same as for Table A1
*#we reduced the number of decimals as requested*

 - Please double check the language of the paper. It has some mixture between the British and American variants of English. For instance, in some places the word 'modeled' is used, and in other you use 'modelled'
*#we double checked language consistency for U.S. English*